



# Understanding Winter Windstorm Predictability over Europe

Lisa Degenhardt[1], Gregor C Leckebusch[1,2], and Adam A Scaife[3,4]

[1]Geography, Earth and Environmental Sciences, University of Birmingham, Birmingham, UK
[2]Institute for Meteorologie, Freie Universitäte Berlin, Berlin, Germany
[3]Hadley Centre for Climate Prediction and Research, Met Office, Exeter, UK
[4]Faculty of Environment, Science and Economy, University of Exeter, Exeter, UK

**Correspondence:** Lisa Degenhardt (LXD943@student.bham.ac.uk)

**Abstract.** Winter windstorms are one of the most damaging meteorological events in the extra-tropics. Their impact on society makes it essential to understand and improve the seasonal forecast of these extreme events. Skilful predictions on a seasonal time scale have been shown in previous studies by investigating hindcasts from various forecast centres. This study aims to connect forecast skill to relevant dynamical factors. Therefore, 10 factors have been selected which are known to influence

either windstorms directly or their synoptic systems, cyclones. These factors are tested with ERA5 and GloSea5 seasonal hindcasts for their relation to windstorm forecast performance.

Following GloSea5 factors' validation contributing to windstorms, the seasonal forecast skill of the factors themselves and the relevance and influence of their forecast quality to windstorm forecast quality is assessed. Factors like mean-sea-level pressure, sea surface temperature, equivalent potential temperature and Eady Growth Rate show coherent results within

these three steps, meaning these factors are skilfully predicted in relevant regions leading to increased forecast skill of winter windstorms. Nevertheless, not all factors show this clear signal of forecast skill improvement for winter windstorms, and this might indicate potential for further model improvements or further understanding to improve seasonal winter windstorm predictions.

## 1 Introduction

Severe winter windstorms are one of the most damaging and loss-bringing events in the extra-tropics, especially for the Eu-

ropean region (MunichRE, 2010). Hence, it is of great scientific interest as well as for stakeholders and the general public to understand these rare extreme events. Studies use various algorithms to identify and track cyclones (Neu et al., 2013). This study aims at understanding an even more extreme event, the surface-near windstorm, which is produced by the strongest of extra-tropical cyclones. Windstorms in this study are thus more related to the direct impacts of a cyclonic system. Leckebusch





et al. (2008) developed an objective tracking algorithm for these strongest wind events. They used a threshold that intentionally
relates to observed losses (Klawa and Ulbrich, 2003) and detects about the top 2% strongest, coherent extreme events in the
extra-tropics. This objective windstorm tracking has been used for multiple different studies in the past, spanning from differ-
ent regions and hazards (Ng and Leckebusch, 2021; Nissen et al., 2013), individual event analysis (Donat et al., 2011b) over
climate (Donat et al., 2011a; Schuster et al., 2019) and seasonal studies (Befort et al., 2019; Renggli et al., 2011; Walz et al.,
2018a; Degenhardt et al., 2022).

Seasonal hindcasts have been investigated in multiple studies for different storm relevant aspects, like the forecast skill of
the North Atlantic Oscillation (NAO; Parker et al., 2019; Athanasiadis et al., 2017; Scaife et al., 2019, 2014), stratospheric con-
ditions (Nie et al., 2019b) or connections between tropical cyclones and extra-tropical storms (Angus and Leckebusch, 2020).
In addition, different regions and events were investigated with respect to their seasonal forecast skill (Dunstone et al., 2018;
Scaife et al., 2017a). For extreme European winter windstorms one of the first studies was published in Renggli et al. (2011)
based on DEMETER (Palmer et al., 2004) and ENSEMBLES (Weisheimer et al., 2009) pilot seasonal hindcasts. More recent
studies investigated later operational systems, like the ECMWF systems (SEAS 3 and 4) and the UK Met Office's GloSea5
(Befort et al., 2019). They found forecast skill in windstorm frequencies and their relation to the large-scale pattern of the NAO.
Following on from this, Degenhardt et al. (2022) found a strong positive and significant signal for windstorm frequency and
(for the first time) intensity. A connection to the three dominant large-scales patterns over Europe showed the NAO, Scandi-
navian Pattern and East-Atlantic Pattern together explain between 60% and 80% of interannual variability of windstorms over
Europe in this seasonal hindcasts, corroborating results from Walz et al. (2018a) based on century-long reanalysis data. This
leads to the motivation for this study, to understand which factors are driving the seasonal winter windstorm prediction skill,
whether as primary or secondary related factors.

Multiple studies investigated dynamical factors influencing cyclone and storm generation and intensification in the past. The
Eady Growth Rate (EGR) parameter (Eady, 1949) is used as a standard measure for baroclinic instability of the atmospheric
flow and is known as a source and intensifying factor for extra-tropical cyclones (Hoskins and Valdes, 1990). Later, i.a. Pinto
et al. (2008) investigated important dynamical factors and their connection to strong cyclones over Europe for future climate
change scenarios, based on previously identified contributors like EGR in the upper troposphere (Hoskins and Hodges, 2002),
upper-troposphere divergence (Ulbrich et al., 2001), the jet stream speed (Kurz, 1990; Hoskins et al., 1983; Shaw et al., 2016)
and the equivalent-potential temperature ($\Theta_e$; Chang et al., 1984). These variables were also used in other studies (Pinto et al.,
2008; Hansen et al., 2019; Walz et al., 2018b; Priestley et al., 2023).

For EGR, this study uses the same diagnostic level of 400hPa as in Pinto et al. (2008) for the upper troposphere but also
700hPa (resulting from 2 available model levels) to diagnose lower troposphere baroclinicity. The location and strength of the
jet stream is important for whether the end of the North Atlantic storm track reaches Europe (Parker et al., 2019). $\Theta_e$ is not
only a measurement for the moisture content in the atmosphere and its static stability but links to the concept of the isentropic
Potential Vorticity (PV; i.a. Hoskins, 2015; Hoskins et al., 1985). Thus, Raymond (1992) could demonstrate that latent heat
release leads to a redistribution of PV, with positive PV tendencies below the level of maximum heating and negative tendencies
above. It is known that the downwards propagation of upper tropospheric positive PV anomaly favours the strengthening of



cyclones (Hoskins et al., 1985; Büeler and Pfahl, 2017). Hence, it is connected to cyclonic systems and can be an indicator
for their strength and location over the North Atlantic. Hoskins et al. (1985) compared different isentropic levels for the PV,
including 350K, which is used in this study as it is a good average representative for the synoptic scales in the troposphere.
They have also connected this concept with Rossby Wave transition. Upper-troposphere Divergence is also part of the equation
for the Rossby Wave Source (RWS), a measure of developing Rossby waves which are transporting cyclones and potentially
transporting predictability from the tropics to the extra-tropics (Beverley et al., 2019; Dunstone et al., 2018; Scaife et al.,
2017b).

Other influencing factors for the generation and intensification of cyclones, are the general environmental conditions which
are thus indirectly connected to windstorms like the sea surface temperature (SST) distribution, SST Gradient and mean sea
level pressure (MSLP) gradient (Shaw et al., 2016). These contributing environmental factors will be called secondary factors
in the following, while factors like EGR or PV which have a direct influence on cyclones and windstorms are called primary
factors. Recently, the SST and the jet stream have been identified as drivers for storm track biases in CMIP6 data (Priestley
et al., 2023). Beyond those generally well established factors, other studies identify the important role of tropical precipitation
as an indicator for European climate predictability (e.g. Scaife et al., 2017b): tropical convective precipitation triggers enhanced
vertical lifting, which again leads to the establishment of Rossby Waves trains impacting on Europe. Further on, Wild et al.
(2015) discovered a dependency of the windstorm frequency over Europe on the temperature gradient between North American
surface temperature anomalies and those of the SST over the western North Atlantic.

This study investigates primary and secondary dynamical factors connected to windstorms in seasonal forecasts from the
UK Met Office, GloSea5 (MacLachlan et al., 2015), and the respective seasonal windstorm forecast skill. This could lead to
better knowledge of the origin of the seasonal forecast skill and hence confidence in real time forecasts.

This study uses a 3-step approach to understand the role of different primary and secondary dynamical factors for the winter
windstorm predictability over Europe.

Step 1: Validation of dynamical factors: Is the observed physical link between factor and storm well represented in the model?

Step 2: Skill of Factors: Is the dynamical factor itself skilfully predicted?

Step 3: Relevance of Factors for Storm forecast skill: Is the forecast skill of windstorms related to the factor's forecast skill or
factor-related "centres of activity"?

The study will first introduce the data sets used in section 2, followed by a description of applied methods in section 3. In
section 4, the results are presented, structured within the above mentioned 3-step-approach. The study finishes with a discussion
and conclusion presented in chapter 5.

## 2 Data

This study investigates the seasonal forecast model of the UK Met Office's Global Seasonal Forecasting System version 5
(GloSea5; MacLachlan et al., 2015), in comparison to ECMWF re-analysis, ERA5 (Hersbach et al., 2019). Both data sets
are used for the consistent time from 1993 to 2016. GloSea5 is a multi-member ensemble model with 4 initialisations per



**Table 1.** Dynamical Factors in connection to storminess, cyclones or windstorms over Europe.

| Factor | | Version | Level | Parameter (ERA5/GloSea5) | Analysis Regions |
|---|---|---|---|---|---|
| Temperature Dipole index | | | | sea surface temperature (6h/6h) | North America (105°-80° W, 38°-55° N) North Atlantic (85°-50° W, 15°-35° N) |
| Sea-Surface Temperature | Secondary | Original | Surface | | Boxes of 10°x10° over North Atlantic |
| | | Gradient | | | |
| | | meridional Gradient | | | |
| Mean Sea-Level Pressure | | Gradient | | mean sea level pressure (6h/6h) | |
| | | Meridional Gradient | | | |
| Total precipitation | | mean | | total precipitation (1h/daily) | |
| | | Only December mean | | | |
| Jet | | Location | 200hPa | u- & v-wind component (6h/12h) | 60°-0° W, 30°-75° N |
| | | Speed | | | |
| Potential Vorticity | Primary | original | 350K | u- & v-wind component, temperature T (6h/12h) | Boxes of 10°x10° over North Atlantic |
| | | Bandpass 2-8d | | | |
| | | Advection | 400hPa | | |
| Equivalent potential Temperature $\Theta_e$ | | | 850hPa | | |
| Eady Growth Rate | | original | 400hPa | u, T, Geopotential (6h/12h) | |
| | | 3d variability | | | |
| | | Bandpass 2-4d | | | |
| | | original | 700hPa | | |
| Divergence | | | 200hPa | u- & v-wind component (6h/12h) | |
| Rossby Wave Source | | | | | |

months (on the $1^{st}$, $9^{th}$, $17^{th}$ & $25^{th}$ of each month) and 7 members per initialisation. Currently, 3 different model versions are available which just differ in small system updates. This study investigates the northern hemisphere winter (December to





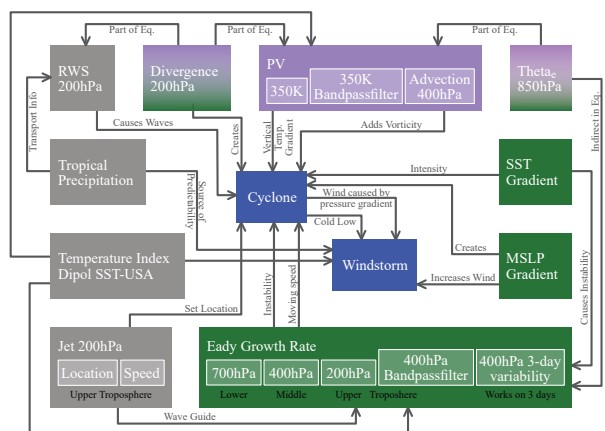

**Figure 1.** Scheme of dynamical factor connection to cyclones and windstorms.

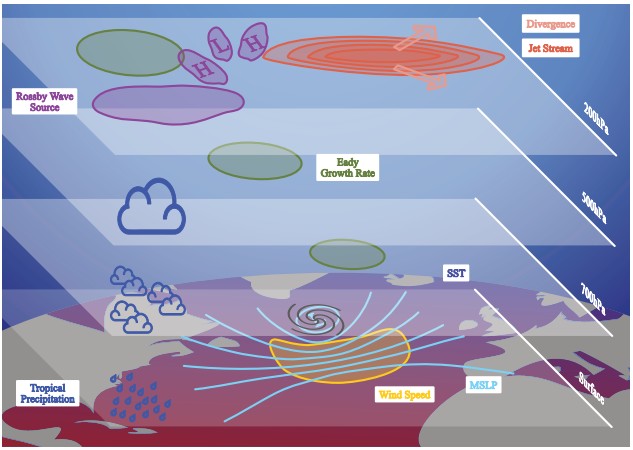

**Figure 2.** Schematic map of location of factors in comparison to an idealised storm system.

February, DJF) and therefore uses initialisation around the 1st of November ($25^{th}$ Oct., $1^{st}$ and $9^{th}$ Nov.). This leads to 63

95   ensemble members (3 system updates x 7 members x 3 initialisations) used here for GloSea5. The seasonal model output has
a spatial resolution of 0.83° longitude x 0.56° latitude. ERA5 is a commonly used re-analysis and provides observation-near
data which are used as reference in this study. The reference data set has a resolution of 0.25x0.25°. Further details of ERA5
can be found in Hersbach et al. (2019). All factors are calculated as described in the method section and variables and levels
used are presented in Tab. 1. The windstorm tracking is based on 10m wind speeds for the calculation (details cf. below). In

100   case of a grid-cell by grid-cell comparison of both data sets, a re-gridding from ERA5 to the spatial resolution of GloSea5 has
been done by a bilinear interpolation using Climate Data Operators (Schulzweida, 2019).





## 3 Method

### 3.1 Storm Tracking

The windstorm analysis is done via an impact based algorithm, developed by Leckebusch et al. (2008). This objective identification and tracking uses a clustered exceedance of the $98^{th}$ percentile of surface wind speeds. These wind clusters are tracked
following a nearest neighbour approach. Only events above a minimum size and duration will be considered: a coherent wind
cluster must persist for at least 48 hours and reach at least a size of 130.000 $km^2$ (cf. details e.g., in Leckebusch et al., 2008).
Consequently, an individual storm track and a grid cell-based footprint of each storm is created. This footprint is used to count
the number of storms over a defined region. The target area in this study in the extended area of the British Isles (-15° to 10° E
& 48° to 60° N). Recently, the authors showed significantly skilful seasonal windstorm predictions for this area (Degenhardt
et al., 2022). The individual windstorm tracks are also used to calculate the track density (used in section 4.3; Kruschke, 2015).

### 3.2 Factors

Dynamical factors are selected by previously known connections to windstorms or cyclones. The selected factors can be
separated into primary and secondary dynamical factors in regard to their connection to windstorms. Hence, primary factors,
like EGR or PV, are dynamical factors which act on a smaller and shorter scale but can influence the cyclone or windstorm
directly/primarily. Secondary factors are acting on a larger and longer scale. These are for example, MSLP gradient or SST, and
they have a more indirect/secondary link to windstorms as they reflect the general state of atmospheric conditions. A summary
of all factors and the way they are used can be found in Table 1. Individual factors are used as seasonal (3 month) averages in
the following analysis.

More details about the different ways of calculating the factors can be found in the appendix. The standard calculations have
been used, e.g., the gradient of MSLP and SST, the jet characteristics (Parker et al., 2019), or the divergence in 200hPa. Other
factors has been calculated followed original studies, like EGR Eady (1949), or PV Hoskins et al. (1985). More unique factors
like Rossby Wave Source (RWS) have been calculated as described in i.a. Beverley et al. (2019) or the Temperature Dipole
used from Wild et al. (2015).

A schematic highlighting the different connections and interactions, is presented in Fig. 1 and 2, illustrating the physical
connectivity between different factors to each other and to cyclones and windstorms in general. The coloured boxes indicate in
which physical view (Quasi-geostrophic Omega- and PV-theory) these factors are included. Fig. 2 is a more exemplary scheme
of an idealised storm-cyclone system, highlighting where the respective factors would be expected to be important. EGR, as
one of the most important factors to strengthen cyclones, is located north-east of the storm centre (at the lowest level) and has a
slope towards northwest with increasing pressure levels. The upper tropospheric baroclinicity (EGR 400hPa) triggers respective
upper-level divergence. The counterpart to this is the SST which influences the low level baroclinicity (EGR 700hPa), which
impacts on the MSLP gradient. The relation of potential predictability of windstorms to convective tropical precipitation (via
vertical lifting triggering a Rossby wave train formation over to the North Atlantic region in higher pressure levels) is tested.



### 3.3  Composite Analysis

135  To understand how and when those factors are influencing the windstorm forecast quality, a composite analysis has been done by separating data sets into two different anomaly categories depending on storm frequency and prediction skill, respectively.

Firstly, a separation is done by the number of storms, thus the seasons' overall activity. The storm counts over the extended area of the British Isles (-15° to 10° E & 48° to 60° N) in ERA5 and each GloSea5 ensemble member are used and separated into 3 categories, the 10 strongest seasons, the 10 weakest seasons and the 3 neutral seasons (10-3-10). A separation into 140  10-3-10-splitting has the aim of still using data sets with at least a decade long duration to achieve representative results, but also to ignore the 3 neutral seasons to reduce the noise. The strong-weak-composites are presented as (member-individual) standardised composite anomalies, to allow for a clear comparison between the ERA5 and GloSea5 data sets.

Secondly, a categorisation with respect to the forecast skill is used: well (bad) forecasted years are identified by using the absolute difference of seasonal storm counts over an individually defined region in GloSea5 and ERA. These categories are 145  built for consistency as well according to the 10-3-10 approach again, i.e., the 10 seasons with the lowest (greatest) absolute difference are used as well (bad) predicted seasons. An example categorisation for individual years can be seen in the appendix (Fig. A1), for ERA5 and GloSea5 ensemble mean windstorm counts in the UK region. Both composite methods are presented as composite anomalies differences, which are tested for significance via a student's t-Test.

### 3.4  Statistical metric of prediction skill

150  All steps of the approach include correlations, here performed using ranked $\tau_b$-Kendall correlations. In more detail, correlation is used in step 1: the verification for the member individual verification (chapter 4.1), in step 2: the skill analysis (chapter 4.2) for the factor individual forecast skill and in step 3: relevance (chapter 4.3) for the storm forecast skill for different data samples. Correlations are a straightforward statistic to use for either relationships between two time series or even forecast skill (e.g., Befort et al., 2019; Athanasiadis et al., 2014; Scaife et al., 2014). Kendall correlation is used because it cannot be assumed 155  that the data are normally distributed. As this study builds up on Degenhardt et al. (2022), the same correlation method is used for a better comparison.

### 4  Results

In the results chapter the focus will be on those 4 factors (2 primary and secondary factors respectively), which highlight the postulated link to forecast skill of winter storms clearly and best, MSLP Gradient, SST, $\Theta_e$ (850hPa), EGR (400hPa). More 160  factors (see Table 1) have been tested within the 3-step-approach but not for all the required links could be clearly identified. Reasons may vary from factor to factor and will be discussed in the discussion section (chapter 5). Additional results for five moderate performing factors can be find in the supplementary material (appendix Fig. A2-A5), EGR (700hPa), MSLP Meridional Gradient, Precipitation, Divergence (200hPa) & PV (350K).

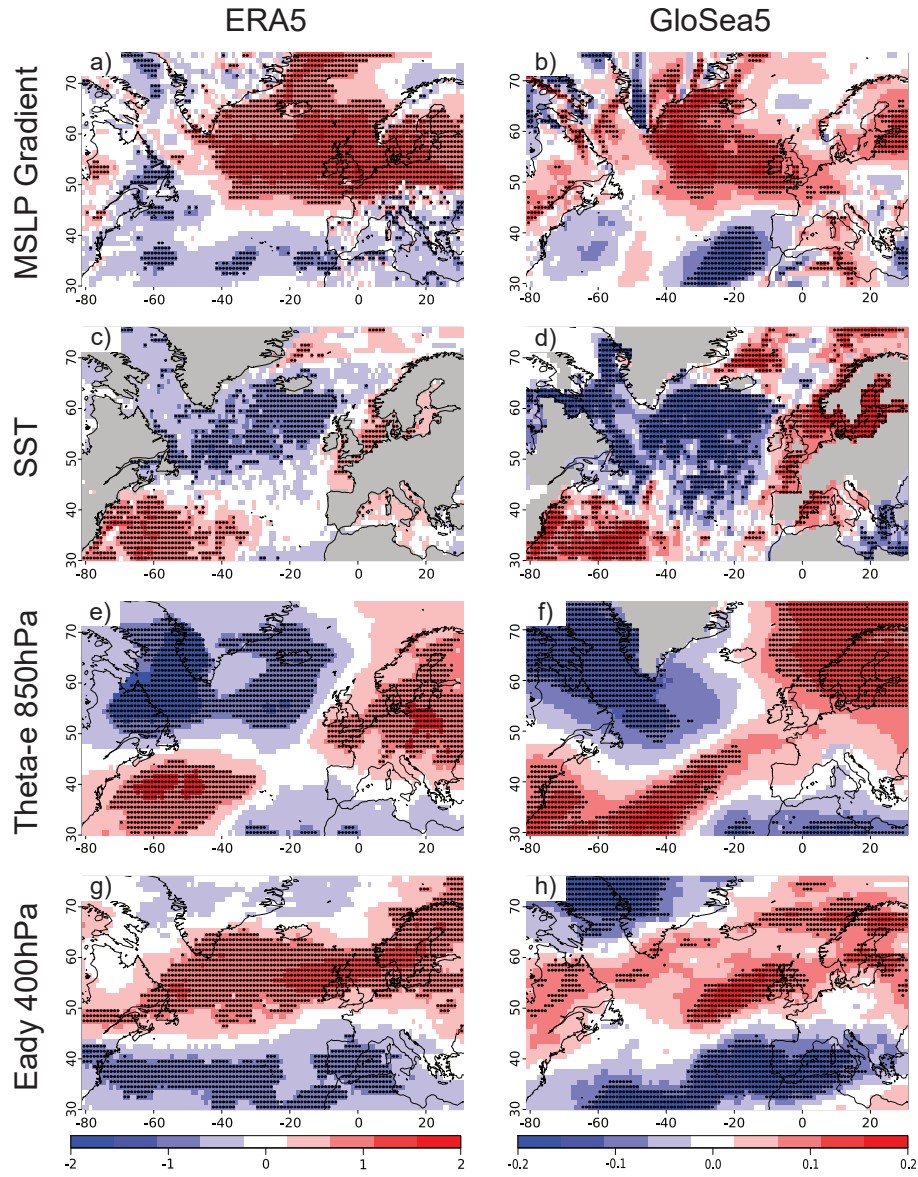

**Figure 3.** Standardised Composite Anomalies of factors for strong vs. weak storm seasons in ERA5 (left column) and GloSea5 mean over all ensemble members (right column): a)&b) MSLP Gradient, c)&d) SST, e)&f) $\Theta_e$, g)&h) EGR; dots shown for differences significant at the 90% level (p=0.9).

## 4.1 Validation of dynamical factors in GloSea5 via anomaly composite analysis

165 Is the physical connection between a causal factor and storm represented in the model as derived from reanalyses? Composite anomalies of the dynamical factors separated into strong and weak storm seasons in the observational and model data are





compared. Standardised composite anomalies for ERA5 and GloSea5 (mean over each ensemble member composite) are used to validate the individual factors on their connection to windstorms in both data sets (Fig. 3; for 4 selected factors discussed in more detail here). The composite anomalies between strong and weak storm seasons give a useful indication of how the factors are connected to windstorms.

For the MSLP-Gradient, it is clearly identified that a stronger storm season is characterised by a stronger MSLP-Gradient over the northern part of the North Atlantic, as expected. This pattern is coherent in observations and the model. The SST-pattern (Fig. 3c & d) shows a clear tripole (positive-negative-positive anomaly) structure over the North Atlantic in ERA5 as well as in GloSea5 (Fig. 3d). The GloSea5 mean signal (mean over all ensemble mean composites) is less strong but still reveals a similar pattern. The three centres of action in the SST composite of ERA5 are reflected as well in the composite pattern of $\Theta_e$. The model mean of composites results in a quadrupole pattern for $\Theta_e$ but with a stronger influence of potential latent heat release over the centre of the North Atlantic than in ERA5. Also, EGR (400hPa) shows a clear and significant pattern over the North Atlantic, with higher baroclinicity in a latitudinal band around 50° N during strong storm seasons over the UK. The secondary factors are known to have a link with cyclones and windstorms, but the former also get influenced by the latter. Nonetheless, the influence of the investigated windstorm systems (max. 2% of days per grid cell) will influence the seasonal average of the factors only marginally.

The appendix includes the composites for more factors (Fig. A2) like EGR (700hPa), MSLP meridional gradient or PV (350K), which are showing in principle similar results as the previous factors, with a strong and coherent increase of the factor itself for stronger storm season over the UK in ERA5 and a good representation of a similar pattern in GloSea5. Nevertheless, precipitation, shows a north-south dipole in ERA5 downstream of the British Isles and Iberian Peninsula, which is less dominant in GloSea5, but also less relevant for windstorm forecasts. As Scaife et al. (2017b) suggest, tropical precipitation is also important for European forecast skill. The model has a strong signal and clear dipole around the equator, revealing more precipitation in the tropics in strong UK-storm seasons.

Composites are categorical separations of data sets, which is useful to clearly identify the difference between two data sub-samples, but the time coherent link between storms and factors is also of great interest, hence a correlation analysis between the factors' time development and windstorm frequency is used for validation. Maps are created to show the correlation link between the windstorm target region (the extended area of the British Isles) and systematic (10°x10° boxes over the whole North Atlantic) regions of the factor over the North Atlantic. Fig. 4 presents the four focused factors as examples, with the remaining in the appendix (Fig. A3).

These results show in more detail the regions where a factor is relevant to windstorms over the extended area of the British Isles (red dotted box in each panel) and how this connection is represented in the different ensemble members (histograms). All factors show in each factor box the correlation has the same sign in ERA5 and GloSea5. Factor regions which are further outside of the storm-related area have some discrepancy, such as the MSLP gradient (Fig. 4a) or PV (Fig. A3e) region over the Mediterranean. In these regions GloSea5 members are not in good agreement with the observational relation. For example, the region around Newfoundland of the EGR (400hPa, Fig. 4d) has a percentile for the ERA5-correlation in the GloSea5-member-

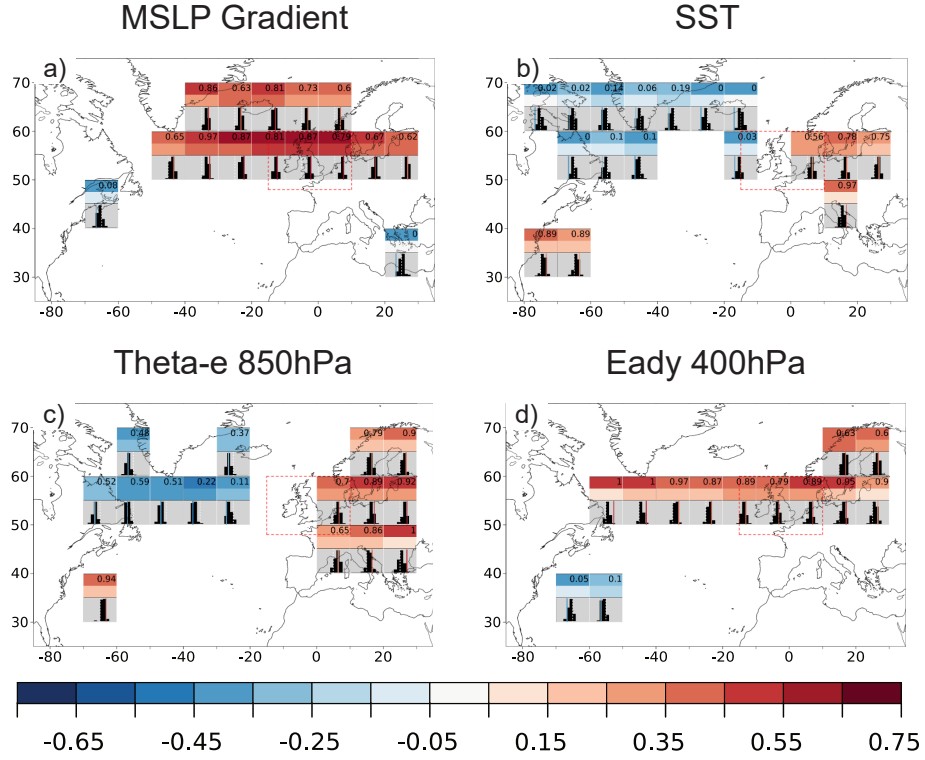

**Figure 4.** Correlation Maps between seasonal storm counts over UK and dynamical factors. Only factors with 95% significant connections in ERA5 are shown. ERA5 connections (1st column), GloSea5 member connection mean (2nd column), GloSea5 individual member connection (below).

distribution of 1, which means the significant correlation in ERA5 is far outside the GloSea5 member correlation distribution and hence statistically different.

## 4.2 Skill of Factors

Is the dynamical factor skilfully predicted? After knowing that relevant factors are well represented in their connection to
windstorms not only from an ensemble mean perspective, but also within individual ensemble members and thus representing a consistent physical development, the next step tests if these factors themselves are well predicted. Thus, this step evaluates how far the necessary ingredients for storm development can be forecasted by the model suite. Thus, in those regions of important connections between factors and windstorms (section 4.1) they should be well predicted to make an influence for the windstorm forecast performance. The Kendall correlation is used to assess the skill of the model's ensemble mean compared
to ERA5. Fig. 5 shows this correlation skill for the main four dynamical factors. MSLP-Gradient has a skilful and coherent region of predictability over the North Atlantic and the British Isles. The SST is overall very well predicted with a small gap upstream of Newfoundland. The same gap but larger and stronger negative correlated is identified as well for $\Theta_e$. EGR (400hPa)

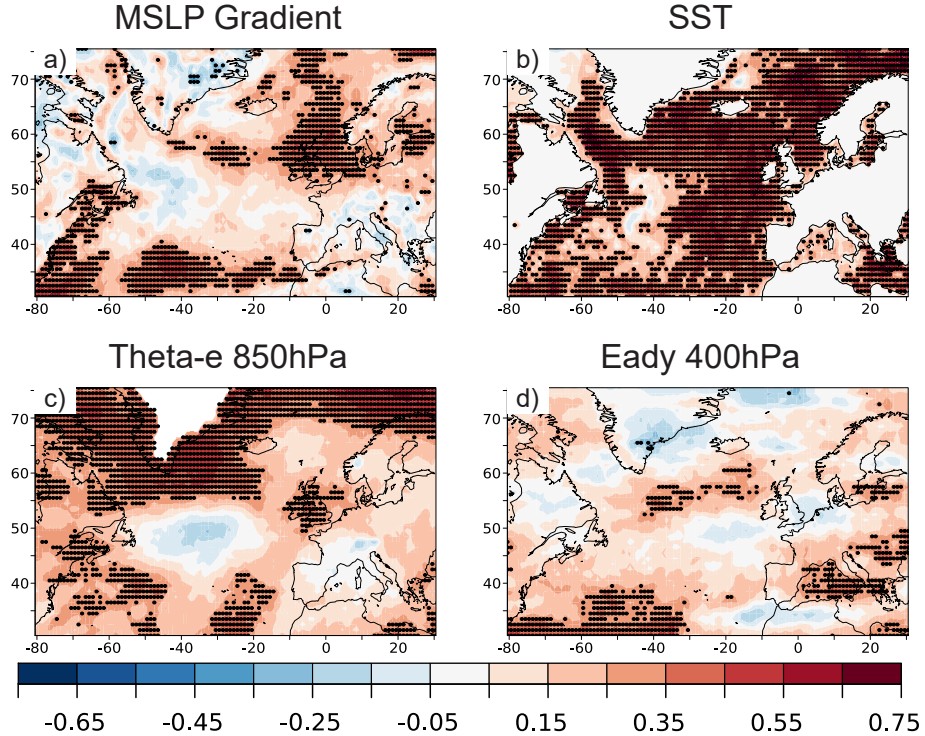

**Figure 5.** Kendall Correlation maps for selected dynamical factors, significance on 95% level marked by a dot.

correlates significant in the region downstream the British Isles which is located north-east of the Atlantic storm track. Beyond the four main factor variables discussed thus far, EGR (700hPa) reveals the same area of skill as 400hPa (cf. appendix Fig. A4). The MSLP meridional gradient shows an extended region of skilful forecasts over the North Atlantic compared to the total gradient, but not the coherent skilful region over the British Isles. Precipitation, divergence and PV 350K all show very little to no skilful prediction close to the target region, the British Isles and Europe. However, precipitation is skilfully predicted in the tropics (cf. appendix Fig. A4) which is the region Scaife et al. (2017b) suggest to be important for European predictability, as this convective precipitation would trigger Rossby Waves which propagate towards the extra-tropics.

## 4.3 Relevance of Factors for Storm forecast skill

Is the storm forecast skill (found by Degenhardt et al., 2022) related to the forecast skill of the factor or the regions that show strong connection to windstorms? To answer this final question, the previous results about factors have been related to windstorm forecast skill. The aim of this step is to find factors and individual regions influencing the seasonal forecast skill of windstorms. Therefore, the storm seasons data has been split into two sub-samples to generate two storm forecast skills depending on each sub-sample. These different characterised storms season sub-samples are separated by two approaches, one





by the factor individual forecast skill (Factor-skill-view, results from chapter 4.2) and one by the centre-of-action from the composite analysis (Process-based-view, results from chapter 4.1).

In more detail:

a) The Factor-skill-view answers the question: "Does the existing factor's forecast skill improve the windstorm forecast?"
Therefore, for the sub-samples of forecast skill, regions are selected that show strong forecast skill for the individual factors, resulting from the approach-step-2: forecast skill (chapter 4.2, Fig. 5). This first view focusses on the regions with already existing and highest factor skill to assess whether the existing positive factor forecast skill in these regions is a source of the existing model's windstorm forecast and a potential improvement. If this is the case, it would mean that the correct prediction of the factor leads to higher storm forecast skill. Thus, the storm seasons are split between well and bad predicted factor seasons.

b) The Process-based-view focuses on the question: "Does areas of strong connection between factor and storm would improve the windstorm forecast?" This second view is using regions that appeared most relevant in the connection between factor and windstorms (centre of action – chapter 4.1, Fig. 3) to create the sub-samples for the different windstorm forecast skills. The aim of this view is to assess if a better prediction of these centres of activity would improve the seasonal windstorm forecast skill. For this, the difference in storm forecast skill (based on correlations) is calculated between sub-samples created
by well and bad predicted factors seasons in the "centre of action"-regions.

Fig. 6 shows differences of the storm skill separated by both approaches, based on successful/bad predictions (factor-skill-view) and on the process-based view, respectively. The region used for separation is marked individually in each panel, some boxes might be out of the mapping area, but all box-details can be found in the appendix (Tab. A1 & A2). For factor MSLP Gradient, three boxes were identified from the factor forecast skill analysis (cf. Fig. 5a) and this correlation difference in Fig. 6a
shows the storm forecast skill for years which are overall well predicted minus storm forecast skill of bad predicted years. It can be concluded that for years in which the MSLP Gradient in these three regions is well predicted, these years show an increase in storm prediction skill over parts of the North Atlantic, British Isles and Scandinavia. In the second view, separated by centres of action in the composite anomalies (Fig. 3a), shows a less strong increase in storm forecast skill for the selected region of MSLP Gradient, but still a slight increase in skill over Scandinavia. This shows the difference between the 2 separations. The
process-based view (centre of action) improves the windstorm forecast skill less and the regions that are already skillful in the factor forecast have more influence on the windstorm forecast skill. As SST was overall well predicted (Fig. 5b) the whole North Atlantic region was used to identify well and bad predicted SST-seasons. When SSTs over the North Atlantic are well predicted, the total storm prediction skill over Europe increases. The Northern European part shows the well predicted years have a significant value on these grid cells but the bad predicted not (indicated by the dotes). The process-based-view for SST
uses the four centres of action defined from the composite-analysis (Fig. 3c) in the North Atlantic. A good forecast in these four centres of action lead to an increase in windstorm forecast skill over Europe as well. The $\Theta_e$ relevance for windstorms is tested by using three regions of skilful $\Theta_e$-forecast (Fig. 5c). When all these three regions are well predicted the windstorm forecast over Europe, especially Scandinavia and East-Europe is increasing (Fig. 6e). This means that the model needs a well predicted $\Theta_e$-value in these three $\Theta_e$-poles to create a skilful or even improved windstorms forecast. As $\Theta_e$ and SST seem to
have a similar link to windstorms (composite patterns in Fig. 3), as potentially higher SSTs result in more convection, hence,



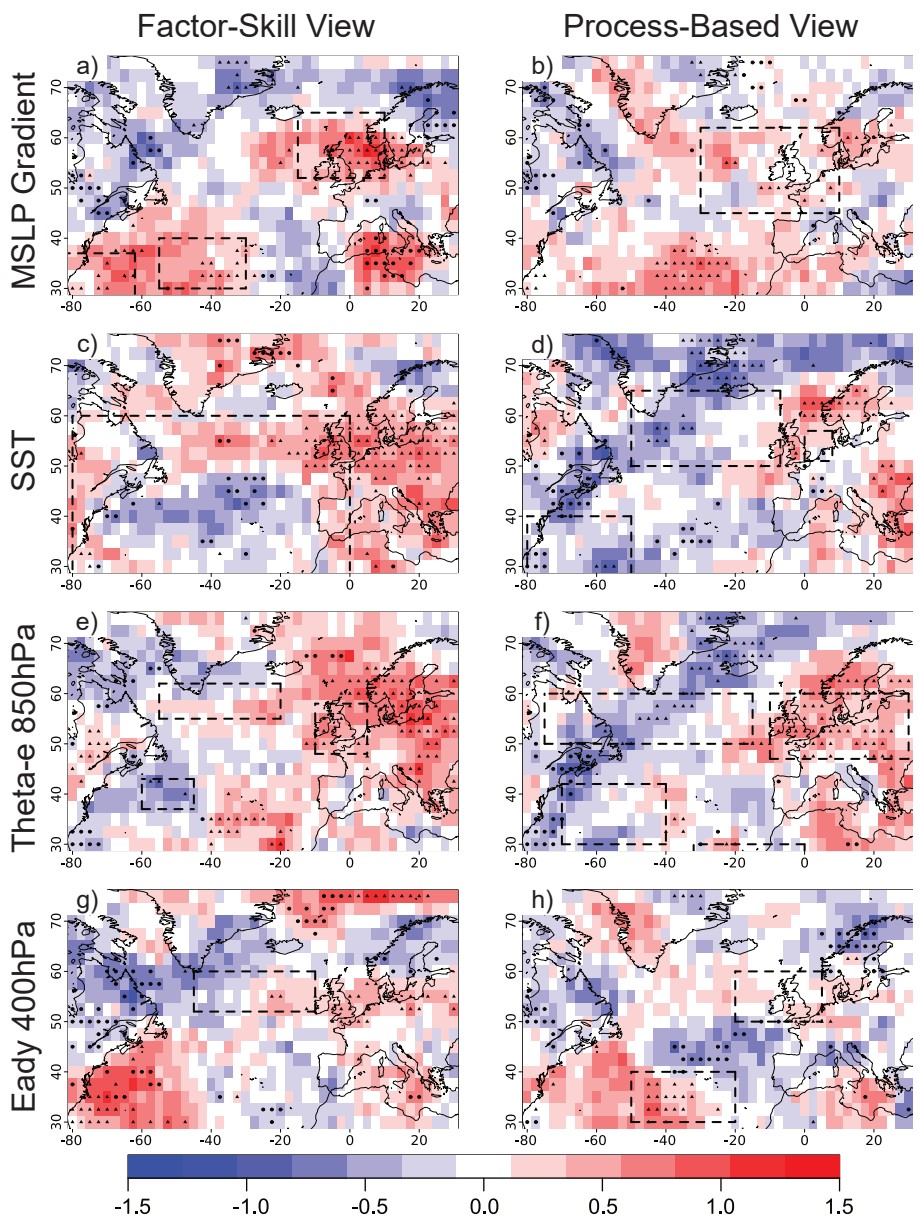

**Figure 6.** Kendall correlation difference between windstorm correlation for (left column) well-bad predicted factor forecast seasons and (right column) centre of actions in composite anomalies. The separation is made by the shown box individually per factor. Dot – bad predicted season was significant, Triangle – well predicted season was significant.

more moisture in 850hPa and a higher $\Theta_e$, these factors show very similar centres of action and the $\Theta_e$ process-based-view has similar four boxes selected as for SST. A good forecast of $\Theta_e$ in these four boxes lead to an overall increase of seasonal windstorm forecast over Europe. This increase is higher and covers a bigger area than the increase by well-predicted SST



regions, might be because $\Theta_e$ is a primary factor, hence, influencing cyclonic systems directly and SST is a secondary factor
which is more an atmospheric state surrounding the cyclonic systems. The relevant signal from the factor-skill-view is not as
strong for EGR in 400hPa (Fig. 6g,h) as for the previous three factors in the same view, but still is a well predicted region
related to an increase of storm forecast skill downstream the box and over the British Isles. As well as the factor-skill-view, the
process-based-view show less increase in windstorm forecast skill for EGR compared to the previous three factors in this view.
The remaining factors can be found in Fig. A5, appendix. EGR in the lower troposphere (700hPa) has two very similar boxes
in both views and hence, almost the same increase in windstorm forecast skill over Europe. Factors like MSLP meridional
Gradient, precipitation and divergence show the skill-dependent selected regions are increasing the windstorm forecast skill
over Europe significantly. The process-based-view is showing increasing signals for factors like precipitation and PV 350K.

## 5 Discussion and Conclusion

This study investigates the connection between primary and secondary atmospheric dynamical factors and the forecast perfor-
mance of seasonal winter windstorm predictions. As skilful seasonal prediction for tracked windstorms (Befort et al., 2019;
Degenhardt et al., 2022) was recently shown, the aim of this study was to further explain the forecast skill. A dependency
of windstorms and windstorm forecast skill on large-scale patterns, like NAO, SCA or EA has previously been established
(Degenhardt et al., 2022). Here, a more in-depth analysis of the mechanics of forecast skill generation is presented and conse-
quently 10 dynamical important factors were selected and tested in multiple settings with respect to their impact on the seasonal
forecast skill of windstorm frequency (see Tab. 1). To reflect on the main contribution of those individual processes to the com-
plex development of extra-tropical cyclones and storms, it has been differentiated between primary (small- and short-scale) or
secondary (large- and long-scale) factors. These factors are investigated in a 3-step approach: first, validation of the relevance
of the factor to winter windstorms. Second, the forecast skill of the individual factor itself on a seasonal scale. And third, the
relevance of the factor's forecast for the overall winter windstorm frequency forecast skill.
The strong link between windstorms and factors seen in the ERA5 composite anomalies of the four focus factors, MSLP
gradient, SST, $\Theta_e$ and EGR (400hPa), are important because these four factors are knowingly the most driving factors for
storm and cyclones (e.g., Pinto et al., 2008). The relation to windstorms for all these important factors is well simulated in
the seasonal forecast suite, GloSea5. The SST shows the known horseshoe anomaly pattern (Nie et al., 2019a) and a clear
connection is identified with a positive SST and $\Theta_e$ signal over Europe (Northern Sea and Baltic Sea): leading to stronger
storm seasons as stronger SSTs may enhance $\Theta_e$, leading to more baroclinic instability e.g., in the lower troposphere in favour
of baroclinic wave development and thus for windstorms. The lower tropospheric EGR (700hPa) agrees with this concept in
ERA5, as the stronger EGR (700hPa) reaches over the North Atlantic until central Europe but lacks in spatial dimension in
GloSea5. The SST composites in GloSea5 show similar three centre of action (positive - east of America, negative – south of
Iceland and positive – North Sea), but a more extended negative SST composite anomaly in GloSea5 further south over the
North Atlantic is in line with the recently found SST bias south of Greenland in CMIP6 models causing a bias in cyclone tracks
(Priestley et al., 2023). The $\Theta_e$ composite anomalies of GloSea5 show a slightly different pattern over the North Atlantic with





a more extended positive signal reaching from south-west to north-east than in ERA5. This is in line with the results from the factor precipitation, where in GloSea5 the North Atlantic precipitation is simulated further west. Studies like Fink et al. (2009) and Pinto et al. (2008) investigated storms from a Lagrangian perspective, but some of their characteristics can also be seen in the here presented Eulerian view. E.g., the dry pole in the north-west of the Atlantic is in line with studies like Fink et al. (2009), which show the general atmospheric state around an extreme cyclone and that a strong cyclone leaves dry air behind. The composites of EGR (400hPa) in ERA5 and GloSea5 show a strong link of EGR just downstream the target area (extended region of British Isles). Especially the pattern of GloSea5 is in line with the knowledge, that EGR affects strong cyclones in a west-east band through their centre (Pinto et al., 2008) and the cyclone centre is located north of the windstorm field (cf. Leckebusch et al., 2008), which explains the strong EGR influence north of the North Atlantic windstorm track.

With mostly agreeing physical connection between windstorms and individual factors within the observational and model data these connections may enhance model forecast performance when the individual factors are well forecast themselves. The individual forecast skills of these factors show high and significant skill in windstorm relevant regions over the North Atlantic but also some gaps. The forecast skill of the upper tropospheric EGR is significant at the north-easterly end of the Atlantic storm track, which is an important area for intensify strong cyclones before making landfall in Europe. Even the forecast skill of the MSLP gradient, SST and $\Theta_e$ show significant skill around the British Isles, but the area around 50° N and 40° to 50° W is a gap for these factors. This reduction in forecast skill may link to previous studies e.g. Scaife et al. (2011) which identify large SST biases in model data.

After the factors have been verified of having the same physical link in observations and models and the model shows forecast skill for important regions of the factor, the third step is connecting the factor performance to windstorm forecast skill. For all main factors it is found that increased forecast skill of relevant factors in relevant regions is increasing the forecast skill of winter windstorms over the British Isles. The process-based-view, sub-sampling based on the centre of action from the composite analysis (step 1), is less conclusive, but especially for SST and $\Theta_e$ the four centres of action help increasing the windstorm prediction over Europe when these regions are well predicted.

The overall conclusion from this three-step approach leads to a well-represented connection between the four focused physical factors and winter windstorm forecast skill. All four factors (MSLP gradient, SST, $\Theta_e$ & EGR 400hPa) show an agreement in the physical link, as composite analysis and in the stricter correlation-maps, suggesting the model does include the physical link overall correctly. For all four factors the model provides positive forecast skill within relevant regions, means the model performance for the individual factor is positive and well predicted seasons in these regions, supporting skilful windstorm forecasts.

In addition, the further investigated factors (cf. appendix) show similar results. Well predicted regions of precipitation and divergence over the tropics and sub-tropics are having a positive influence on the storm predictability over Europe. For precipitation this is in line with Scaife et al. (2017b), which found that tropical Atlantic precipitation as an influencing factor for European predictability of atmospheric patterns. Further crucial factors (not shown) in this study were e.g., the Rossby Wave Source (RWS), SST gradient (total and meridional component) or the North-America/North-Atlantic temperature gradient identified by Wild et al. (2015). For the factor RWS no clear pattern or relation was identified. The ERA5 composite is



very scattered, but the GloSea5 mean shows at least a pattern agreeing with the conceptional idea of the tropical North Atlantic precipitation triggering convective rising which triggers the RWS further North (Scaife et al., 2017b). A similar scattered result is resulting for all approach steps for the SST gradients. The temperature dipole from Wild et al. (2015) has been tested, as a

connection between North American surface temperature and North Atlantic sea surface temperature anomalies are linked to windstorms over Europe. But the results in this study are not conclusive, probably because the storm target region is different in both studies.

This study concludes that the existing windstorm forecast skill in GloSea5 can be explained by different dynamical atmospheric factors which are primarily or secondary connected to cyclones and windstorms. Thus, the model is predicting the

winter storm season well for the correct reasons, increasing confidence in forecasts. Secondary and large-scale factors like the MSLP gradient or SST have a strong relation to windstorms in the observational and model data sets. Their individual seasonal forecast skill is high and seasons which are well predicted have a positive influence on windstorm forecasts. The same is found for primary factors like $\Theta_e$ in 850hPa and EGR in the upper (400hPa) troposphere. This approach results in new understanding of dynamical factors and covers multiple perspectives, which give new knowledge where the windstorms forecast skill might

originate and where additional efforts, beside the also for windstorms existing signal-to-noise paradox (Degenhardt et al., 2022), are needed to potentially improve windstorm forecast skill over the downstream end of the North-Atlantic storm track.

*Author contributions.* Idea and concept of the study are from all authors, calculations and manuscript writing are done by LD with the support of GCL and AAS.

*Competing interests.* The authors have not disclosed any competing interests.

*Acknowledgements.* L. Degenhardt has been supported by the UK's Natural Environment Research Council (NERC) DTP2 CENTA2 grant (NE/S007350/1). The authors thank the Copernicus Climate Change Service (https://cds.climate.copernicus.eu/cdsapp#!/home). This data collection includes, the GloSea5 forecast model and the ERA5 reanalysis data, and is available to freely use. A. A. Scaife received support from the United Kingdom Public Weather Service (https://www.metoffice.gov.uk/about-us/what/pws) and the Met Office Hadley Centre Climate Programme funded by the United Kingdom Government Department of Business, Energy and Industrial Strategy (BEIS) and the

Department of Environment, Food, and Rural Affairs (Defra). The computations described in this paper were performed using the University of Birmingham's BlueBEAR HPC service, which provides a High Performance Computing service to the University's research community (see http://www.birmingham.ac.uk/bear for more details).



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

## Appendix A

This part of the Appendix include additional information about the method and calculation for the dynamical factors, hence for chapter 3.2.

MSLP and SST represent more general information about the environmental conditions. Their respective gradients are
calculated using the NCL (NCAR Command Language) implemented function (grad_latlon_cfd) and compute the absolute value of the gradient vectors. The Climate Data Operator (CDO; Schulzweida, 2019) has an implemented function (uv2dv) to calculate from both wind components (u & v) the respective wind divergence. For the calculation of the Rossby Wave Source (RWS) the python package windspharm (Dawson, 2016) was used as an example script from GitHub. This script is based on the RWS equation used e.g., by Beverley et al. (2019); Dunstone et al. (2018). Studies like Parker et al. (2019) investigated
the jet stream on its seasonal predictability and connection to the NAO. This study follows their calculation of jet location and speed but for 200hPa rather than 850hPa. The jet is defined over a 9-day running mean of the zonal average of the wind, both only the u-component or the total wind was tested. The jet location is defined here as the latitude at which the maximum wind (respectively u or total wind) is found and as jet speed the respective wind is used. An investigation from Wild et al. (2015) analysed how temperature anomalies over North America and the North Atlantic can influence the winter windstorm
season over Europe. They created a Temperature-Dipole index which uses surface temperature at 2 regions, one over North America (105° - 80° W, 38° - 55° N) and one over the western North Atlantic (85° - 50° W, 15° - 35° N). The difference of the respective anomalies creates the so-called temperature index. The PV (Hoskins, 2015; Hoskins et al., 1985) is calculated using two implemented NCL-functions (pot_vort_isobaric & int2p_n_Wrap). Therefore first, the pressure level data are used to calculate PV on pressure levels and secondly these values are interpolated onto $\Theta$-levels. The 350K-level is later used in this
study. The PV Advection is calculated from the pressure-level data and then advected by both (u & v) wind components. $\Theta_e$ as an individual factor on 850hPa (Chang et al., 1984), is calculated with the NCL-function, pot_temp_equiv. The Eady Growth Rate (EGR) is calculated with an implemented NCL function (eady_growth_rate) which uses the 2-layer approach. This means whenever it is referred to EGR at 400hPa it is calculated by using data from 300hPa and 500hPa and for EGR at ∼700hPa it is 500 & 850hPa. Both, PV and EGR are additionally analysed in this study after an additional post-processing, a bandpass filter.
This bandpass filter was run with an R implemented function using the Butterworth filter (Butterworth, 1930), with a filter characteristics of 2 to 8 days for PV and 2 to 4 days for EGR. The filter was performed for each GloSea5 member individually.





Because of data storage and computational times, the filtering was only executed for a region -100° to 40° E and 30° to 75° N. The total precipitation is used as in Scaife et al. (2017b) to investigate the link between tropical precipitation and predictability of European climate conditions, like geopotential height. To be not restricted on the four used tropical regions used in Scaife

et al. (2017b) and for a better comparison to the other used factors, the seasonal precipitation mean is investigated on grid-cell level.

This part of the Appendix includes the results for the remaining tested dynamical factors. Therefore, it belongs to the Result chapter 4.

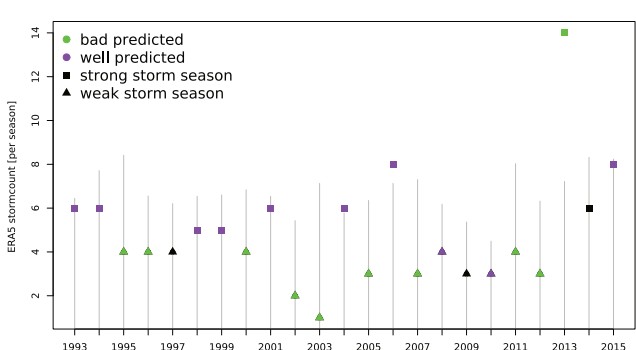

**Figure A1.** ERA5 UK storm counts as dots and GloSea5 ensemble mean counts as bars. Bad predicted seasons (green), well predicted seasons (purple), weak ERA5 seasons (triangles) and strong ERA5 seasons (squares).







**Figure A2.** As Fig. 3 for remaining primary and secondary factors.

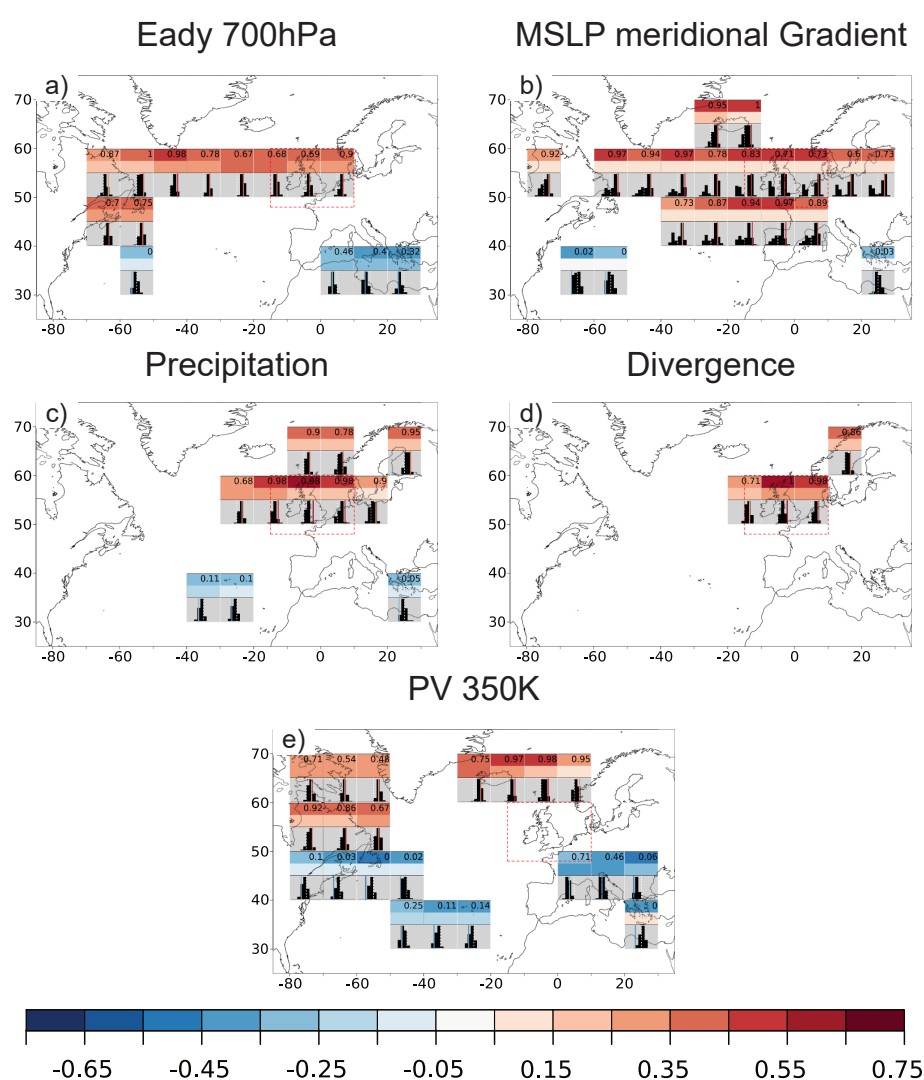

**Figure A3.** As Fig. 4 for remaining primary and secondary factors.



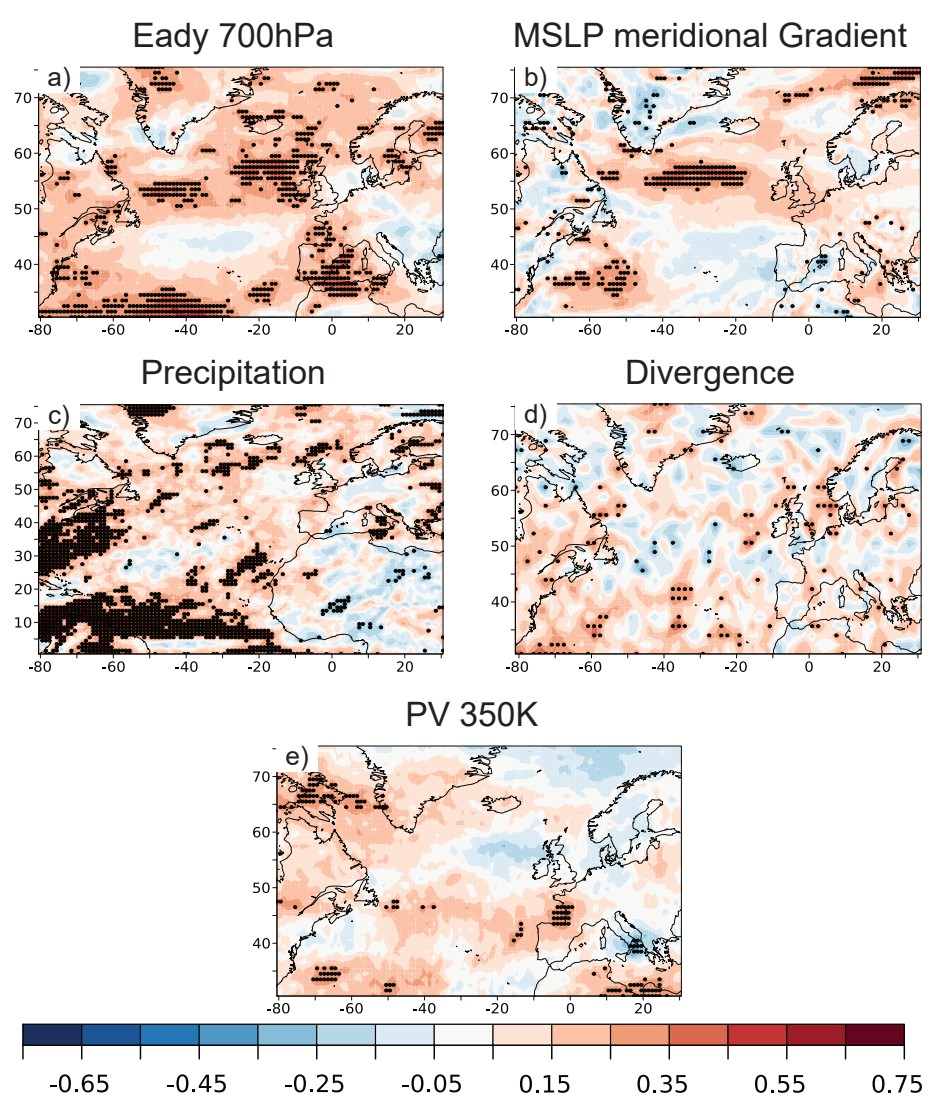

**Figure A4.** As Fig. 5 for remaining primary and secondary factors.





**Figure A5.** As Fig. 6 for remaining primary and secondary factors.





**Table A1.** Tested skilful regions of factor forecast skill.

| Factor | Box 1 | Box 2 | Box 3 | Boxmean |
|---|---|---|---|---|
| Mean Sea-Level Pressure Gradient | -15° - 10° E 52° - 65° N | -55° - -30° E 30° - 40° N | -85° - -62° E 27° - 37° N | Box 1-3 |
| Sea-Surface Temperature | -35° - -5° E 40° - 64° N | -80° - -45° E 20° - 35° N | -80° - 0° E 10° - 60° N | Box 1&2 |
| Equivalent Potential Temperature $\Theta_e$ | -10° - 5° E 48° - 58° N | -60° - -45° E 37° - 43° N | -55° - -20° E 55° - 62° N | Box 1-3 |
| Eady Growth Rate 400hPa | -45° - -10° E 52° - 60° N | | | |
| Eady Growth Rate 700hPa | -25° - 0° E 50° - 60° N | | | |
| Mean Sea-Level Pressure Meridional Gradient | -60° - 5° E 50° - 60° N | -80° - -50° E 10° - 30° N | | Box 1&2 |
| Total Precipitation | -85° - -15° E 5° - 20° N | -90° - -55° E 20° - 45° N | | Box 1&2 |
| Divergence | -90° - -65° E 20° - 30° N | | | |
| Potential Vorticity 350K | -30° - -5° E 52° - 59° N | -10° - 5° E 42° - 48° N | -30° - -10° E 12° - 24° N | Box 1-3 |



**Table A2.** Tested relevant regions of ERA5 & GloSea5 composite anomalies.

| Factor | Box 1 | Box 2 | Box 3 | Box 4 | Boxmean |
|---|---|---|---|---|---|
| Mean Sea-Level Pressure Gradient | -40° - 0° E 30° - 40° N | -30° - 10° E 45° - 62° N | -40° - 0° E 15° - 30° N | | Box 1-3 |
| Sea-Surface Temperature | -80° - -50° E 27° - 40° N | -50° - -7° E 50° - 65° N | 0° - 8° E 51° - 57° N | -20° - -10° E 21° - 27° N | Box 1-4 |
| Equivalent Potential Temperature $\Theta_e$ | -70° - -40° E 30° - 42° N | -32° - 0° E 25° - 30° N | -10° - 30° E 47° - 60° N | -75° - -15° E 50° - 60° N | Box 1-4 |
| Eady Growth Rate 400hPa | -50° - -20° E 30° - 40° N | -20° - 5° E 50° - 60° N | | | Box 1&2 |
| Eady Growth Rate 700hPa | -70° - 10° E 25° - 35° N | -30° - 0° E 50° - 60° N | | | Box 1&2 |
| Mean Sea-Level Pressure Meridional Gradient | -70° - -30° E 30° - 40° N | -30° - 10° E 45° - 57° N | -40° - 0° E 15° - 30° N | | Box 1-3 |
| Total Precipitation | -37° - 0° E 30° - 40° N | -25° - 10° E 50° - 62° N | | | Box 1&2 |
| Divergence | -40° - -7° E 42° - 27° N | -15° - 7° E 45° - 63° N | | | Box 1&2 |
| Potential Vorticity 350K | -5° - 20° E 45° - 60° N | -80° - -52° E 15° - 23° N | | | Box 1&2 |