# Peer review of "Understanding Winter Windstorm Predictability over Europe"

_Weather and Climate Dynamics, 2023_

## Author Comment (AC1)

Anonymous Referee #1, 18 Jul 2023

Overall comments:
The aim of this study is to investigate potential sources of European winter windstorm predictability. The overall methodology appears to make sense, but I found the article hard to read (it really needs a thorough proof-read as there are many poorly-worded or grammatically incorrect sentences; in some cases to the extent that it wasn't really clear what you mean).

In particular, I didn't fully understand the "process-based view" method, so please could you review the description of this to make sure it is clear.

Some of the figures need a bit of work to make them clearer – in particular I found the schematics in Fig 1 and 2 confusing (and also some of the text on them was too small and hard to read – not sure if this was in the conversion to pdf or an issue with the actual figures).
[LD] The current figure setting is for one column, we could increase the figure size inside the manuscript to a two-column figure, but also try to adjust the font within the figures. Another option is to remove figure 1 completely from the manuscript and move figure 2 to the appendix, this would lead to only explanations of the physical connections as text within the manuscript.

Specific comments:
Abstract: The second paragraph of the abstract seems rather vague. Please could this be rewritten – perhaps if you look at the conclusions section and summarise the key results in a more solid way.
[LD] The second paragraph can be adjusted, rephrased and clarified with more details. The 2nd abstract paragraph could be simplified by using numbering for the 3 investigation steps and give a clear statement for each investigation step as summary.

L38: what do you mean by "signal" here?
[LD] This refers to the correlation skill score used in the mentioned publications. This can be changed to a more precise word ("skill")

L 50: can you say what the equivalent-potential temperature actually is (either here or later) as I, and presumably other readers, am not familiar with this quantity.
[LD] More explanation of Theta_e can be added to the manuscript, especially the introduction. E.g. "$\Theta e$ is a parameter that describes the temperature of a fully dried air parcel dry-adiabatically lowered onto a reference level, usually 1000hPa (Bolton, 1980)."

L105: do you think your method will detect sting-jet storms? These storms have some of the strongest and most damaging surface winds (e.g. the Great storm of 1987) but are relatively short-lived and affect a relatively small area compared with

other types of storms where the strong winds are due to e.g. the cold-conveyor belt winds. I wonder if your tracking method would capture these events and if so, whether they might have different dynamical drivers than other types of storms. This may be beyond the scope of your study but definitely worth considering/mentioning.

[LD] The used algorithm is not designed to detect sting jets or cyclones. The algorithm is an impact-based tracking, that is locating synoptic-scale exceedances of wind-speed. The authors are aware of the interest in sting jets, but this is not part of this study as sting jets would only be possible to detect on smaller scales.

L122 check reference format, should be in brackets

[LD] Thanks for pointing this out, this will been changed.

Throughout: you seem to use "i.a." instead of "i.e." or "e.g."

[LD] "i.a." stands for inter alia, but we can change it throughout to "e.g."

Figure 1: Caption should be "schematic" rather than "scheme". Some of the text within the figure is very small.

[LD] Please see comment above. Figures were set to one column figures, but can be changed to 2-column, so they appear bigger. And font can be adjusted.

Figure 1: In general, I didn't really understand this schematic and I think more description is needed in the figure caption and/or text. In particular:

- What do the different coloured boxes mean? In the text it says "The coloured boxes indicate in which physical view (Quasi-geostrophic Omega- and PV-theory) these factors are included" but it doesn't say anywhere what each individual colour corresponds to (this should be in the figure caption).
- What are your definitions of "cyclone" and "windstorm" that means these are considered as separate things? And for instance, why is tropical precipitation labelled as a "source of predictability" for windstorm but not for cyclone?
- What is the PV 350K bandpassfilter and how is it different from 350K?

[LD] We would think about removing this quite complex figure from the manuscript

Figure 2: Text is small/fuzzy and the text on the diagonal lines is squashed and hard to read. There is some description of this figure in the text but it wasn't clear to me how some of the text corresponded to what is in the schematic and there are many aspects of the schematic which are not explained. For example what are the small rain-cloud shapes which appear to be at the surface? I appreciate that there are lots of complex processes involved and it would be useful to summarise them in some sort of schematic, but this particular schematic does not seem to show them very clearly and requires some work.

[LD] These suggestions can easily be change by increasing the figure size, the font size and add more details into the text. (e.g. "The upper tropospheric baroclinicity

(EGR 400hPa) triggers respective upper-level divergence (peach) and hence, creates the jet stream (orange). The counterpart to this is the SST (ocean colour) which influences the low-level baroclinicity (EGR 700hPa), which impacts the MSLP gradient (light blue) and hence, the wind speed (yellow). Another process related to the potential predictability of windstorms is caused by convective tropical precipitation (dark blue) via vertical lifting, triggering a Rossby wave train (purple) formation to the North Atlantic region in higher altitudes.")

L140: Do you do this for each ensemble member separately? So for each ensemble member you do the 10-3-10 classification, which means that when you then look at composites they are based on different sets of years for the different members and for the observations? Could you make this clearer? Also if this is the correct interpretation, could you comment on how much similarity/difference there is between the ensemble members and the obs: do the same years get picked out by most of the ensemble members as high/low storm count years? Are they consistent with the obs? This could have implications for the interpretation of the results in terms of the influencing factors.
[LD] We can add more explanation to the manuscript. Yes, we are doing each separation individually per member, as we really want to have the physics of seasons with many storms against less storms in each realization. We can add more explanation to the manuscript (like "The separation is done individually per model ensemble member to ensure that each composite compares strong vs weak storm seasons internally.")

L140: what do you mean by "at least a decade long duration"? A decade is a period of 10 years, but I think you mean 10 years of data?
[LD] Yes, this is a very good suggestions and we could take the phrasing of "10 years of data".

L144: is this the total absolute difference of individual members or of the ensemble mean?
[LD] This is based on the ensemble mean, as the forecast skill is also based on that. (could be changed to "using the absolute difference of the respective seasonally averaged factor over an individually defined region in the GloSea5 ensemble mean and ERA")

L150 Please can you give a brief description of what the ranked Tb-Kendall correlation is
[LD] A more detailed description of the used measure can be added, like "Kendall correlation is a similar measure to the commonly used Person's correlation but investigates ranked time series and is less subject to normally distributed data."

L151: "chapter" should be "section"
[LD] Can be changed throughout the manuscript to "section".

L165: This is strangely written: what do you mean by "represented in the model as derived from reanalysis"? Do you mean something like "Does the model represent the same physical connections between causal factors as the reanalysis"?
[LD] Yes, the suggestion of the reviewer is right and we would like to adopted this phrasing.

Figure 3: It's not clear to me what has been plotted here. Is it the respective quantity in the strong storm seasons minus the weak storm seasons, at each grid point? Why is the magnitude in the GloSea5 mean so much smaller than in ERA5?
[LD] Yes, it is the respective quantity as difference between strong vs. weak storm seasons. As this has been calculated per each ensemble member, it is possible that some members have the same strength of the link and others don't. Hence, an averaged over all members to present the average difference for GloSea5 could result in a smaller difference than ERA5.

Figure 4: I found this figure very confusing! Please add more description in the figure caption. Where you say "connection" I think you mean "correlation" and where you say 1st/2nd column you mean row not column. The bar graphs don't have a scale so I'm not sure how useful they really are. And I thought the colours corresponded to correlation values but then the numbers in the boxes don't seem to correspond. I'm also a bit confused by the interpretation of the figure and the conclusion you draw from it that the connection between the factors and windstorms is well represented in ensemble members as well as ensemble mean (L204/5).
[LD] The number that has been referred to is not the correlation values. The colour refers to the ERA5 correlation, the number refers to the percentile of the ERA5 correlation within the GloSea5 member distribution. This can be made more clear in the figure caption. (E.g. "The coloured-red line is the ERA5 correlation value within the GloSea5 member distribution, and the number represents the percentile of ERA5 in that distribution.")

L213: do you mean upstream of the British Isles rather than downstream?
[LD] that is correct, I am sorry for the mix-up.

Figure 6: I didn't find it clear what this is showing, please can you explain more clearly in the caption or text. My understanding: you're showing the Kendall correlation difference in windstorm forecast at each grid point, between seasons in which the various factors or processes are well- and poorly-forecast. And the boxes show the regions that you're interested in for the factors/processes. But how do you combine the boxes to determine if the season is well or poorly forecast? What if it's well forecast in one box region but not another?
[LD] I spatially average the respective quantity in the shown boxes and then use the same process with absolute difference and compare it between ERA5 and GloSea5.

This could be a potential additional sentence for clarification:

"For both views, the selected regions (which can be multiple) are spatially averaged, and well- and bad-predicted seasons are detected by the absolute difference between the resulting ERA5- and GloSea5-time series in the used regions. The regions for the factor-skill-view are all skilful for the respective factor. For the process-based view, this is not a criterion."

Figure 6: the dots and triangles are too small to be able to differentiate between them. And it's also not clear to me what they mean. What does it mean for a well predicted year or a badly predicted year to be significant?
[LD] This is supposed to be a significance measure. A dot/triangle is added if either the first or the second parameter in the difference is significant but not both.

L255 (and similar description/interpretation of results): does your definition of well forecast or badly forecast necessarily mean that the factor is well forecast in all the boxes? Where you have several boxes could the quantity be very well forecast in just one or two of the boxes but not in the others?
[LD] I have checked each box individually and the combination of all of them, for calculation details, see above.

L317: it might also be worth saying that the skill is increased over other parts of Europe – NW Europe, SW Europe?
[LD] This could be easily added to the text. Along these lines "t has been found that all main factors increase the forecast skill of winter windstorms over the British Isles and the North Sea by increasing the forecast skill of relevant factors in storm-relevant regions. SST and Θe additionally improve the windstorm forecast skill over Central Europe and Southern Scandinavia."

---

## Author Comment (AC2)

Review of wcd-2023-12

The seasonal forecast skill of European windstorms is investigated in this study. In particular, the dynamical factors that potentially drive the known skill in seasonal forecasts of storms are assessed. It is shown that for four key dynamical drivers of cyclones: their representation in the seasonal forecast model is similar to ERA5, the seasonal forecast of storms is correlated to the dynamical factors in various upstream regions, and well forecast storm seasons are correlated with well forecast dynamical factors. This topic is definitely of interest to the community and fits well within the scope of the journal. The methods used seem appropriate (though they are not always well explained) and the results are interesting. However, I found the paper to be very hard to follow. There are many poorly-worded sentences and poorly-described figures. I would think the manuscript would be suitable for publication after a thorough proof read and strong edit for clarity and completeness. More in depth comments are included below.

[LD] Thank you for the comment and following suggestions.

Major comments:

1. Clarity of writing.

I found much of the text hard to follow. The article would benefit from a thorough rewrite to draw out the main aims, results and implications of the study, which are currently being lost within the somewhat unclear text/structure and confusingly worded/long sentences. For example, the introduction contains several paragraphs about seasonal forecasts and the dynamics of extratropical cyclones, but they are not well linked with each other or related to the aims of the study (which are not really mentioned until the end of the introduction). I have listed some sentences that were unclear to me below, but it is not an exhaustive list and I recommend the entire manuscript be checked for clarity. Also, several sections begin with a question, which is presumably the question that the section aims to address. The questions need to be properly introduced and answered if they are to be included, it reads as draft-like in its current format.

[LD] The manuscript can be checked in total for more clarity with a focus on clear and shorter sentences, more description in Figure captions.

The methods need to be more clearly explained as well. In particular, it is not clear what exactly is being shown in the Figures or how it is calculated. Most of the results show correlations but there is no information on exactly what is being correlated. The results certainly could not be reproduced with the information that is currently included.

[LD] Correlations are always Kendall correlations which can be stated in the method section, and the figure captions can be adjusted with more information to clarify what is correlated in each figure.

Unclear sentences:

L7: "Following Glosea5 factors' validation contributing to windstorms"

[LD] This part of the abstract was suggested to rephrase by Reviewer I as well and can be clarified with numbering the investigation steps: "Following GloSea5 factors' are (1) validated on the physical connections to windstorms, (2) investigated on the seasonal forecast skill of the factors themselves, and (3) assessed on the relevance and influence of their forecast quality to windstorm forecast quality."

L23: "Windstorms in this study are thus more related to the direct impacts of a cyclonic system". More related than what?

[LD] Further details could clarify this sentence "Windstorms in this study are thus more related to the direct impacts of a cyclonic system rather than just the low-pressure systems."

L59: "Hence, it is connected to cyclonic systems and can be an indicator for their strength and location over the North Atlantic".

[LD] This sentence can be adjusted and clarified.

L190: "but the time coherent link between storms and factors is also of great interest, hence a correlation analysis between the factors' time development and windstorm frequency is used for validation"

[LD] This sentence is explaining Fig. 4. We could write it in shorter sentences to make it clearer. "Composites are categorical separations of data sets, which are useful for identifying the difference between two data sub-samples clearly. A time-coherent link between storms and factors is also of great interest. Hence, a correlation analysis between the factors' time development (as time series) and windstorm frequency (as storm counts) is used for additional validation (see Fig. 4)."

L204: "After knowing that relevant factors are well represented in their connection to windstorms not only from an ensemble mean perspective, but also within individual ensemble members and thus representing a consistent physical development, the next step tests if these factors themselves are well predicted."

[LD] Here as well, with writing it in shorter sentences, we can try to make it clearer. "The previous results summarise that relevant factors are well represented in their connection to windstorms. This had been shown for an ensemble mean perspective (with composites, Fig. 3) but also within individual ensemble members (correlations per member, Fig. 4). Thus, the GloSea5 model represents a consistent physical development between respective factors and windstorms. The next step tests if these factors themselves are well predicted."

L207: "Thus, in those regions of important connections between factors and windstorms (section 4.1) they should be well predicted to make an influence for the windstorm forecast performance."

[LD] Maybe reduce the sentence to make it clearer: "The storm-relevant regions (section 4.1) should be well predicted to have a positive influence on the windstorm forecast performance."

L306: "With mostly agreeing physical connection between windstorms and individual factors within the observational and model data these connections may enhance model forecast performance when the individual factors are well forecast themselves".

[LD] We see that is was a confusing sentence. By breaking it down into 2 shorter sentences, we hope the message gets clear: "The physical connections between windstorms and individual factors within the model data mostly agree with the connections in the observational data. These connections may enhance model forecast performance when the individual factors are well forecasted in the storm-relevant regions."

L323: "For all four factors the model provides positive forecast skill within relevant regions, means the model performance for the individual factor is positive and well predicted seasons in these regions, supporting skilful windstorm forecasts."

[LD] Same as above, we hope a paragraph with shorter sentence will make this statement clear: "The model provides positive forecast skill within relevant regions for all four factors, which means the model performance for the individual factor is positive. The final investigation step shows that well-predicted seasons of the factors in the relevant regions support skilful windstorm forecasts."

L333: "A similar scattered result is resulting for all approach steps for the SST gradients."

[LD] We think the double result is not well chosen, so we could change to "result is seen for all"

L344: "which give new knowledge where the windstorms forecast skill might originate and where additional efforts, beside the also for windstorms existing signal-to-noise paradox"

[LD] Hopefully a rephrasing along these lines help for clarify ", which implies new knowledge about where the windstorm forecast skill might originate. This also reveals areas for additional efforts needed to potentially improve windstorm forecast skill over the downstream end of the North-Atlantic storm track, alongside the also for windstorms existing signal-to-noise paradox"

2. The dynamical factors.

Much of the analysis focuses on four of the dynamical factors that are deemed most influential for cyclone development, yet there are 20 (by my count) that are included in Table 1 and Figure 1. I wonder if it is necessary to include all the factors in Table 1 and Figure 1 as you do not really mention them in the text (the coloured boxes in Figure 1 are not defined either).

[LD] Figure 1 will be adjusted with a legend.

The schematic in Figure 2 is also not properly described. I would recommend removing the Figures and Table and simply listing the predictors you chose to analyse in the study. If you do keep all the predictors in the manuscript then there should be a much more thorough description of what each means and how they relate to cyclone development (though I'm not sure what the point of this would be as the majority of the predictors are not included in the main text).

[LD] We understand this comment, but wanted to show, that we have not only checked the four focused factors but a bigger list of potential factors. Table 1 could be moved to the appendix or reduced to only the focused factors. Fig. 1 and 2 are

supposed to show the different levels of interaction between factors and cyclones/windstorms. We think about removing Figure 1 from the manuscript and move Figure 2 into the appendix, so reduce complexity of this part. But still more details can be explained in the text.

There is also no clear explanation on how the four included predictors are chosen (you say they "highlight the postulated link to winter storms clearly and best"). What metric is used to determine this? This information would potentially be more beneficial to show than the schematics.
[LD] We choose to have 2 primary and 2 secondary factors in the paper, but more in the appendix, to not overload the manuscript. The way of choosing was a step-by-step investigation which factor show a clear result throughout all investigation steps.

3. Selection of good and bad forecasts.

I am somewhat confused on how you separate good and bad forecasts for the results presented in Figure 6. In section 3.3 it says you separate forecast years into good and bad by comparing their storm counts to that in ERA. But then in section 4.3 it says you separate them into good and bad by considering the skill of the forecast factors (though it is not clear exactly what you mean by this). I have a number of concerns about the approach regardless:

-Are you considering at all the temporal aspect of forecast skill or if the skill is actually related to wind storms? If you are just comparing the mean values of the factors in the different regions across the entire forecast then I'm not sure you can relate this purely to windstorms. For example, you might have a low value of MSLP gradient in the different regions that is well predicted and which is associated with a good prediction of a reduced number of storms. Therefore the skill may increase over the UK but not in relation to storms. (I could be misunderstanding what is shown in the plot.)
[LD] I am not fully sure I understand this comment. We are only using the factor forecast skill to separate the seasons. And then investigate if these different sub-samples of season have different characteristics in storm forecast skill. We believe there are more ways of doing this but this would exceed the aim of the manuscript.

-To me, a more intuitive approach would be to consider the factor skill in the regions when a storm is identified. Then you could show that when a storm is in the forecast and the factor regions are well predicted, the storm is well predicted over the British Isles, and vice versa. You have the tracks for the storms so this should be feasible.
[LD] This sounds like a reasonable approach, but is not the idea we wanted to follow with this study. Our aim wasn't to look at individual storm tracks or events, we wanted to look at a general state of the atmosphere during the windstorm seasons and depending if it is a stormy or not stormy season.

-Do you require the skill to be good in all the factor regions? If so, have you tested if a particular region is most important. I.e. does the forecast skill over the UK increase more if the factor is well predicted in a particular region?
[LD] the regions have been tested individually itself (see table in the appendix), the figures shown in Fig. 6 are only the ones with the highest change (between well and badly predicted factor) in correlation. Meaning if multiple boxes are shown all boxes are considered in this particular panel, but the rest has been tested as well, but was less significant.

-Is the difference in the left and right columns of figure 6 just that the regions used to define the good and bad forecast skill are different? But the method is the same apart from that?
[LD] yes exactly, we could add something like "…separation by the Factor-skill-view (left column) and the Process-based-view (right column). The separation is based on spatial averages over the shown boxes from Fig. 5 for the left column and Fig. 3 for the right column, …" in the Figure caption.

-Have you tested different metrics of forecast skill? The results presented may be sensitive to the metric you use to determine forecast skill (please state clearly what this is). It would be good to try other metrics and compare results.
[LD] This study is based on a previous study of the authors, where we show also different metrics for the forecast skill. Kendall correlation was the most intuitive to use and too understand, but we agree, that the results can be dependent on the chosen metrics.

Minor comments:
L45: the Eady Growth Rate parameter is not itself a source and intensifying factor for extratropical cyclones. Strong baroclinicity is (i.e. what high values of the EGR parameter represent). So this sentence needs rephrasing.
[LD] This sentence should be correct by changing and to which "The Eady Growth Rate (EGR) parameter (Eady, 1949) is used as a standard measure for baroclinic instability of the atmospheric flow  which is known as a source and intensifying factor for extra-tropical cyclones (Hoskins and Valdes, 1990)."
L50: "These variables were also used in other studies". Other studies about what? How were the variables used? Some additional context is needed here.
[LD] We will write more context to these studies.
L77: "This could lead". This seems vague and weak. You could say something like "the aim of this study is to better understand…". Or something similar.
[LD] Thanks for this suggestions, something along these lines can be used to change the sentence
L115: Do you mean local PV? Remote PV anomalies can influence cyclones via action at a distance.
[LD] We are not specify local or remote, as some investigation/method steps consider a spatial distance of the factor to the target storms (e.g. Fig. 3, Fig. 4 or Fig.

 They have the target of UK storms but investigate the factor (here PV) for the whole North Atlantic.

L118: Did you test if your results are sensitive to the averaging length? 3 months seems quite a long time period to average for dynamical factors relating to cyclones.

[LD] We agree with that. The aim of the study was to look into the seasonal time scale. Some factors (EGR & PV) have been tested as bandpass filtered version to take into account that they are important on a smaller time scale, but their results were very scattered and not conclusive. This is included in the Method-section in the appendix.

L150: Please define what is meant by tau_b Kendall correlations. How are they calculated?

[LD] This can be clarified with the following sentence "Kendall correlation is a similar measure to the commonly used Person's correlation but investigates ranked time series and is less subject to normally distributed data." We could rather not add too many equations of established statistics, they can be found in the mentioned citation.

Throughout: use of chapter instead of section!

[LD] I hope this reviewer means the same as the first reviewer. We decided to go with section for the whole manuscript and changed it to be consistent.

Figure 3: Are there compensating errors here? I.e. do the strong storm seasons look similar in GloSea and ERA, as well as the weak storm seasons, their differences might look similar for the wrong reasons.

[LD] That is a good point, this can be checked internally and mentioned in the manuscript.

L187: would a dipole suggest a shift in precipitation location rather than an overall increase?

[LD] That is true, thanks for the comment, we will change the "more precipitation" to "shifted precipitation"

Figure 4: unclear exactly what you are correlating here? You correlate the number of storms with what metric of the dynamical factor (mean in the boxes?).

[LD] Caption can be adjusted with an additional explanation like "Correlation Maps between seasonal storm counts over the UK and dynamical factors (averaged in 10x10◦ regions)"

Figure 4: can you include the correlation values for GloSea as well? This would allow for easier comparison than comparing redness/blueness. The histograms are very small here as well.

[LD] The value inside the ERA5 row is not the correlation value.

Figure 5: again, it is not clear what is being correlated here.

[LD] Can be adjusted by something like "Kendall Correlation maps for selected dynamical factors between ERA5 and GloSea5 per grid cell"

L231: what aspect of factor skill are you referring to here? The mean value of the factor in the region? Temporal evolution? Please state explicitly.

[LD] This is the explanation of one approach for the investigation step #3. This sentence is about the regions selected to create subsamples of the storm data. By

adding a few more words, we hope this sentence is clearer, like "that show coherent regions of skilful forecasts for the individual factors".

L276: the aim of the study here should be more clearly stated in the introduction [LD] This has been stated in the introduction, but we could extend the introduction sentence like this "These skilful storm forecasts found in seasonal hindcasts lead to the motivation for this study. This study aims to understand which dynamical factors drive the seasonal winter windstorm prediction skill, whether as primary or secondary related factors."

Technical corrections:

L2: the seasonal forecast of —> seasonal forecasts of

L5: I'm not sure if ERA5 and GloSea5 should be included in the abstract without defining them. Perhaps change to "a reanalysis product and a seasonal forecast system".

L10: What three steps? [LD] by adjusting the abstract we added numbers for the 3 investigation-steps

L21: use rare or extreme. Do not need both.

L26: remove "from" before "different regions and hazards".

L44: "investigated" —> "have investigated".

L46: I'm not sure if i.a. is right here?

L54: Need to define theta_e before you use it.

L73: "Further on". Further on than what? The study you refer to is from 2015 which is earlier than those mentioned previously. [LD] rephrased with "another factor discovered by …"

L90: GloSea5 is defined earlier, though not fully? [LD] Added earlier around line 35

L109: in —> is

L127: "exemplary" means very good. I do not think that is what you mean here. [LD] I am not a native speaker, hence, I have to google and trust my online dictionary, but I find "exemplary" as the adjective for an example

L143: bad —> badly

L155: up on —> upon

L174: less strong —> stronger? [LD] less strong = weaker, have changed this

L189: is —> are

L198: outside —> upstream? [LD] no "outside", not only upstream but everywhere where it is not storm-relevant

Figure 4 caption: column —> row

L206: using "Thus" to start two sentences in a row, should be changed.

L212: upstream —> downstream? [LD] no, upstream is with the flow, meaning east of something in the midlatitudes

L213: downstream —> upstream? [LD] no, same reason as previous comment, vice versa.

L235: Does —> do

L235: remove "would" after storm.

L264: might be theta_e —> which might be theta_e

L265: is SST an atmospheric state? [LD] changed to "global"

L328: which —> who

[LD] Thanks for the small corrections, will be changed.

---

## Author Response (AR1)

**Combined Answer to the Review Process**

The authors want to thank the reviewers and editor for the suggestions and time they spent supporting this manuscript.

The revised manuscript includes now all recommended changes in line with the more detailed answers we already provided in the discussion.

Nevertheless, three more substantial modifications have been implemented to increase the clarity of the writing and to convey our approach in a clearer way, as recommended by the reviewers and editor:

- We decided to delete the separation of factors into "primary" and "secondary" factors. We initially thought this might help in structuring the results, but it seems this was an unnecessary complexity.
- Similarly, to clarify and to streamline the methodic approach, we decided to delete the "mind map" (previous Fig. 1), to reduce the factor table to the core information and to move the total table and the 3-D scheme (previous Fig. 2) to the appendix. This will reduce the complexity of the method chapter and we think will increase the clarity of our approach.
- Additionally, as recommended, we introduced changes in the explanation for the prosses-based view. We shortened and modified the explanation which hopefully leads to a better understanding. We also adjusted the composite explanation in the method-section slightly to make this a clearer extinction between the different composites.

---

## Author Response (AR2)

Review of Degenhardt et al Understanding Winter Windstorm Predictability – revised submission

In general the paper is much clearer than the initial submission. I still found the last part (section 4.3, and corresponding figures 4 and A6) a bit hard to follow. Otherwise, I would be happy for the paper to be published after addressing the following comments.

L130 Do you really mean that these processes in the cyclone creates the jet stream? Isn't the jet stream a larger-scale phenomenon? Do you mean this strengthens the jet stream?

[LD] This is more an explanation of the general process in the atmosphere. This should not say, that the cyclones are creating the jet stream. We change the wording accordingly to avoid any misunderstanding.

L130 what do you mean by "ocean colour"?

[LD] This was misleading. We wanted to point out, that the SST is represented as the colour of the sea areas.

L188 I didn't understand where you were talking about here, in terms of the differences in precip – do you mean upstream rather than downstream? And if you do mean downstream would these differences really have an impact on storm forecast skill? Wouldn't the precipitation differences here be a result, rather than a cause, of the imperfect storm forecasts?

[LD] We meant downstream, sorry for the mix-up.

Fig 2: it's hard to see the coloured lines in some of the distribution panels – not sure how you could make them clearer, maybe increase the line width? The blues are a bit easier to see than the reds. Does the colour of the ERA5 line change according to the value (not sure if it does or if it just looks different depending on context)? If so maybe don't do this, and choose a single colour for the ERA5 value that is easiest to see.

[LD] yes, the line was the same colour as the ERA5 correlation value. We tried a different colour and hope this provides better visible throughout the figure now.

L206: I don't understand this sentence. From the histogram, it looks like the ERA5 correlation is within the range of the GloSea5 correlation distribution – what is the evidence for these being statistically different?

[LD] The histogram in the back only shows a little bar for that category. You need to be careful, as the histogram counts in ranges to create the bars, as the ERA5 value is an exact value. It is the case that the value from GloSea5 within that bar is left of the ERA5 value, hence the ERA5 value is outside the GloSea5-distribution. We tried to add more explanations to make this clearer without prolonging the sentence too much.

L220: do you mean downstream not upstream?

[LD] Yes, sorry, we changed accordingly.

L226: although the skill does not seem to be present actually at the equator, only from about 5N.

[LD] True, according to Scaife et al. (2017), their Fig. 2, the Atlantic skill is highest at the 5° northern border of their selected box, so very much in agreement.

Figure 4: I found this figure a bit confusing – I think the caption could be clearer about what exactly is shown here. Reading the text (paragraph beginning on L256) helped me to understand a bit more what the plots are showing but I think you need to be clearer in the caption.

[LD] With restructuring the caption and adding more information, we hope this is now clearer from the caption itself.

L264: It's not clear to me how you define this. Isn't the SST factor based on the tripole structure of the SSTs? So how do you define the SST skill over the whole N Atlantic?

[LD] In the factor-skill-view we take the whole North Atlantic, as the forecast skill of SST is significant over most of the North Atlantic. For the Process-based-view you are right, and we are only taking the tripole structure.

L359: Can you elaborate on how this relates to the signal-to-noise paradox? In particular:

Is the s-to-n paradox seen in GloSea5 for windstorms? (has this been shown?)

How do your results relate to the s-to-n paradox – i.e. how could your results help to explain this, or help to reduce the s-to-n issue?

[LD] We added a more explanatory sentence about the s-to-n paradox. And yes, the stn-paradox is also found in windstorms, as we showed in Degenhardt et al. (2022).

General point: "bad predicted" is incorrect: it should be either "badly predicted" or "poorly predicted"

[LD] Many thanks, yes, we changed accordingly throughout the manuscript

wcd-2013-12 second review: Understanding Winter Windstorm Predictability over Europe

I appreciate the effort the authors have made following the first round of reviews. Improvements have definitely been made in terms of clarity and readability. The results are now better summarised, particularly in the discussion section. I do however still think that more needs to be done in places to improve the written English in terms of both grammar and clarity. I believe that if these comments can be addressed the manuscript will be suitable for publication in WCD, as the results are interesting and add to the growing body of work on seasonal forecast skill.

Please find below my comments.

Major comments:

1. There are still many occasions where sentences are improperly structured and/or lack clarity. Given that at least one of the coauthors is a native English speaker, I am surprised that this is the case given the comments from both reviewers in the first round of reviews. I think that for this paper to be published it still requires a thorough proof read and edit for clarity and readability. I have suggested some changes in the comments below to some of the most obvious places where the writing can be improved, but I encourage the authors to check for others.

> [LD] The native speaker of the author team went through the whole manuscript again and proof-read the manuscript in great detail.

2. Have you compared how the windstorm forecast changes if you used random regional boxes (I.e. not related to forecast skill or factor relevance)? It would make your results stronger if the increases you find are very different than for random regions.

> [LD] Many thanks for this comment. Yes, we did test with different boxes, but all dependend on forecast skill or the physical connection postulated. Each of the results does show slightly different increase in windstorm forecast skill but here we present the combination of boxes with the strongest difference to identify the most important and significant links.

> Unfortunately, further random testing cannot be performed right now in full detail, as this is work from a PhD Thesis which is already completed and there is no current access to the data and script. If this is would be a requirement before publication, we would unfortunately, need more time to regain access to University computer systems to test this idea. Nevertheless, we are confident that the way we addressed this problem is well-suited to provide robust and comprehensive results to this problem.

Minor comments:

L24: "studies use various…". Is this sentence needed here?

> [LD] This sentence introduced the idea of tracking storms instead of using a general state or pattern of the atmosphere. But we can see that this was unnecessary, so we deleted it.

L130: smaller and shorter scale than what?

> [LD] we added "than other tested factors"

Fig.2: why not include the actual correlation value for GloSea in the box that just is coloured at the moment. For completeness these should be included.

> [LD] The value in the ERA5 box is not the correlation value corresponding to the colour. It is the percentile value where the correlation of ERA5 is in comparison to the GloSea5 member distribution (see captions)

Fig.2: are the regions that the factors are included only based on ERA5? Are they the same if based on GloSea?

> [LD] Yes, the selected boxes shown are only separated by ERA5. All GloSea5 mean correlations are slightly lower than ERA5, hence, none other box would be added.

Discussion of Figure 3: can you explain/postulate why there is a gap in correlations downstream of Newfoundland?

> [LD] We link this gap to previous studies that find an SST bias in the model for that region. This is included in the Discussion section.

Figure 4: make it very clear that the regions for the left panel are selected based on the results from Figure 3 and the regions for the right panel are based on the results from Figure 1 (I think this is correct).

> [LD] Yes, this is correct. We moved this higher up in the caption and also added it into the Figure itself.

L305: mention that the process-based view skill change is weaker and regionally dependent.

> [LD] We added more information here: ", but with a weaker change and mostly only for skill over Scandinavia"

L324: It would be useful to summarise here what is shown in Figure 4, i.e. the factor-skill view improves windstorm forecasts more than the process-based view, and what this means.

> [LD] Done, we added a short summary here.

Examples of unclear sentences:

L82: sentence beginning "Another factor discovered by .."

L186: sentence beginning "All investigation steps.."

L271: sentence beginning "This first view focuses on…"

L293: sentence beginning "For factor MSLP gradient…"

L337: sentence beginning "The strong link…"

> [LD] We  simplified these sentences by shortening them or separate into multiple shorter sentences. The whole manuscript has been carefully read for clarity and proof read again.

Suggested technical/writing corrections:

L2: remove "the" from "the seasonal forecasts"

L6 and elsewhere: re-analysis —> reanalysis

L38: add "the" before "DEMETER"

L44: this seasonal hindcasts —> these seasonal hindcasts

L58 and elsewhere: EGR —> Ehe EGR

L62: measurement —> measure

L64: could demonstrate —> demonstrated

L66: upper tropospheric PV anomaly —> an upper-tropospheric PV anomaly

L74: Other influencing factors for —> Other factors influencing

L81: to impact —> that impact

L93: factor's —> factors'

L100: the seasonal forecast of —> seasonal forecasts from

L128: Dynamical factors are selected by previously known connections to windstorms or cyclones. —> The dynamical factors included here are selected based on their known connections to windstorms or cyclones.

L143: are you trying to say that the schematic is very good? If not, "exemplary" is not the correct word here and I suggest changing "an exemplary schematic" to "a schematic". If you are, add justification for why the schematic is so good.

L143: highlighting —> which highlights

L153: those —> these

L157: members —> member

L161: vs —> versus

L177: the verification for the member individual verification —> verification of individual members

L185: in the results section —> in this section

L190: moderate —> moderately

Caption Fig.1: on climatology —> with respect to the climatology

GloSea5 mean over all ensemble members —> over the mean of all GloSea5 ensemble members

dots shown —> dots are shown

L196: are compared —> are compared in Figure 1

L204: weaker —> stronger?

> [LD] weaker is right, as the scale of the GloSea5 composites is smaller than from the ERA5 composites, hence the signal is weaker.

L215: downstream —> upstream. Definition of upstream "in the opposite direction from that in which a stream flows; nearer to the source". The Atlantic is upstream of the UK, the UK is downstream of the Atlantic. You use upstream in the correct way on L251.

L240: had been —> was.

L243: an agreeing —> a similar

L244: how far —> how well?

L248: MSLP gradient —> The MSLP gradient

L250: upstream —> downstream. Definition of downstream "in the direction in which a stream or river flows."

L250: and stronger —> and with a stronger

L252: factors variables —> factors

L278: The regions for this view —> The regions used for this view

L293: For factor MSLP gradient —> For the MSLP gradient factor

L308: is increasing —> increased

L314: add references to the panels you are comparing here to help the reader follow

L341: remove the semicolon

L345: similar three —> three similar

L361: spacial —> spatial

L392: conceptional —> conceptual

> [LD] all technical/writing suggestions have been changed accordingly in the manuscript.

---

## Author Response (AR3)

**Public justification (visible to the public if the article is accepted and published)**:
Dear Lisa Degenhardt and co-authors,

Thank you for submitting the revised version addressing the reviewers' comments.

As noted by the first reviewer in the previous round, having read the new version I still think Figs 4 and A6 need better explanation in the text. As these are key figures of the paper, the message emerging from them should be more clearly outlined. There are also quite a few sentences that can be improved throughout the text.

[LD] Dear Shira Raveh-Rubin,

Many thanks for spending again valuable time on the editorial process of our manuscript. We are thankful for the already made comments and suggestions of the two reviewers and yourself and try to reply in the following as detailed as possible to the additional advice provided by you. We identified several places in the text where we think we could improve the quality of writing. We started with the abstract and made some clarifying language changes. Further changes are marked in the track-changes version of the revised manuscript.

I suggest below some specific changes to improve the readability of the text and figures. Note the line numbers refer to the last author tracked changes version.

[LD] Many thanks for this- much appreciated. We comment point-by-point in our reply.

Fig. 4 and accompanying text: The readability of the figure and its results should be enhanced.

I suggest that before describing the results (in the paragraph in lines 265-271), better guiding of the reader through the figure would be helpful, along these lines

> "i.e., red colours in the left column of Fig. 4 mean that… while in the right column the interpretations of red colours would be… areas with different colours in the left/right columns for the same variable imply that…".

[LD] Many thanks for this. Yes, we added a respective explanation at the end of the mentioned paragraph. We chose this to be placed at the end of this paragraph, as this paragraph firstly describes the Fig. 4 in general and the new details of the figure interpretation would then nicely allow the reader to move on to the description of the results, which are presented in the direct following section.

More specifically, it is unclear from the caption if the difference in skill shown is "well-predicted minus poorly-predicted" forecasts for both approaches (columns), please be specific.

[LD] Many thanks, for this comment. This information has now been added to the figure caption.

The parenthesis and notion "respectively" in line 266 are unclear (I assume it is not successful predictions for the factor-skill view and unsuccessful predictions for the process-based view).

[LD] Many thanks for highlighting this. We double checked and think the sentence is in principle correct. Nevertheless, we increased clarity by re-phrasing to: "The differences in the respective forecast skill of the storm frequency for these two approaches are shown in Fig.4. The left column provides the differences in skill for the factor-based view, the right column the difference in forecast skill for the process-based view."

Also, the dots and triangles are indistinguishable in print (also in Fig. A6), please enhance this in the figures.

[LD] Many thanks for making us aware of this. We added colouring to distinguish dots and triangles better: triangles are now given in green, dots in black colour. We also increased the size of the dots and triangles.

Line 269: should "absolute difference" be replaced by "correlation"?
[LD] Many thanks for highlighting this ambiguity. The phrase "absolute differences" is right here. The categorisation whether a season is poor or well predicted is based on the absolute differences of the individual factor. The region mean is used as time series and each season individually subtrahend (ERA5 minus GloSea5 ensemble mean). This is described in the method section roughly L139ff (tracked version).

Table 1: please clarify the direction of MSLP gradient (or if the maximal magnitude of MSLP gradient is considered at each grid point), and why the u,v components are mentioned for calculating theta_e (and mixing ratios are not)
[LD] The MSLP-gradient is given as its  absolute value, thus without a direction, we added this to the table. You are right with Theta_e, this is a typo that probably occurred by copying cells from other factors. The Theta_e is calculated with the specific humidity and temperature. Thank you very much for making us aware of this.

Lines 12-13: merge these two sentences in the abstract "These factors are skillfully predicted in storm-relevant regions. And this skill leads to increased forecast skill of winter windstorms over Europe".
[LD] Done, please cf. further language changes to the abstract as mentioned above.

Line 28: delete "about"
[LD] We replaced this with "ca.", as the tracking is based on the 98[th] percentile of the local wind speed and the existing of a spatial coherent cluster of a minimum size of this exceedance and with a minimum duration. Thus, only if all exceedances of the local 98[th] percentile are occurring in large enough clusters it would be exactly 2%. In reality the value may thus be a bit lower, depending on how many small-scale clusters with only short duration occur.

Line 52, 66: change to "upper-tropospheric"
[LD] Done

Line 75: delete "on"
[LD] Done

Line 128: add "the" before "original"
[LD] Done

Line 158: delete "is"
[LD] Done

Line 201: Here you changed "downstream" to "upstream", which is the opposite of the reply to the reviewer's question. Please clarify.
[LD] Many thanks for pointing this out. As reviewer 2 commented this should read "upstream", as the dipole pattern is in the upstream direction of the general westerly flow in this region. Thus, as the dipole pattern is west of the UK, "upstream" should be correct.

Lines 209-210: delete repetitive "over the North Atlantic".
[LD] Done

Line 211: Seems better not to start a new paragraph here.
[LD] Done

Fig. 2 caption: add "histograms" before "below". Change "scales" to "scaled"
[LD] Done

Line 252-3: this sentence is not very clear, please rephrase and place it after the next sentence to help with readability.
[LD] We newly structured this paragraph and shortened the sentences for more clarity.

Line 259/262: is centre of action = centre of activity? If so, choose one and be consistent throughout.
[LD] Done

Line 283: change the first "the" -> "that"
[LD] Done

Line 284: one -> ones
[LD] Done

Line 291: "result in more convection": this is not shown directly so either delete this statement or frame it as a possible explanation.
[LD] We have framed this sentence more as an explanation to make it clearer that the convection part is not shown.

Fig. A2 caption: bad -> badly
[LD] Done

Fig. A4: change from red/blue to yellow lines here as in Fig. 2.
[LD] Done

Check for misplaced capital letters throughout the manuscript (e.g., MSLP Gradient, Divergence and others)
[LD] Done

Additional private note (visible to authors and reviewers only):
Dear reviewers, thank you for the second round of constructive suggestions!
Shira